SPECIAL ISSUE
LIFELONG DEVELOPMENT

# *egal-1* and microtubules promote regeneration polarity in planarians

Yochabed Miliard[1,2,‡], Shannon Moreno[1,2,‡], Lauren E. Cote[1,2,*,‡] and Peter W. Reddien[1,2,3,§]

## ABSTRACT

A central problem in regeneration is how the identity of new tissues is specified. A classic example is the head-versus-tail regeneration decision in planarians. *notum* is wound induced at anterior-facing planarian wounds, where it triggers head regeneration through inhibition of canonical Wnt signaling. This represents the earliest known asymmetric regeneration step between anterior- and posterior-facing wounds. Wound-induced *notum* is specific to longitudinal (anterior-posterior-axis oriented) muscle cells, suggesting these fibers might harbor polarity harnessed for the head-tail regeneration decision. The processes that occur within longitudinal muscle after injury for preferential *notum* activation at anterior-facing wounds are poorly understood. We utilized single-cell RNA sequencing to identify multiple wound-induced genes in longitudinal muscle cells and identified processes required for wound-induced *notum* asymmetry. Egalitarian-like-1 (Egal-1) is wound induced in longitudinal muscle and has some domain similarity with *Drosophila* Egalitarian, which facilitates asymmetric RNA localization. Both *egal-1* RNAi animals and animals with destabilized microtubules (via colchicine or nocodazole treatment) show ectopic *notum* expression at posterior-facing wounds. We suggest that Egal-1 and microtubules are together required for longitudinal muscle fibers to promote planarian regeneration polarity.

KEY WORDS: Regeneration, Planaria, Wound signaling, Microtubules, Stem cells, Neoblasts, Blastema, Egalitarian, Muscle

## INTRODUCTION

One central challenge in animal regeneration is the initiation of a program that is tailored to the identity of missing cell types. This involves patterning processes that specify the identity of tissues to be made and that regulate how they are organized. A classic context for studying this problem is the choice to regenerate a head or a tail at transverse amputation planes in planarians (Reddien and Sánchez Alvarado, 2004). Planarians are flatworms capable of regenerating any missing body region. Small body fragments in some planarian species occasionally make errors, resulting in two-headed or two-tailed

[1]Whitehead Institute for Biomedical Research, Cambridge, MA 02142, USA. [2]Department of Biology, Massachusetts Institute of Technology, Cambridge, MA 02139, USA. [3]Howard Hughes Medical Institute, Massachusetts Institute of Technology, Cambridge, MA 02139, USA.
[‡]These authors contributed equally to this work. [*]Present address: Department of Biology, Stanford University, Stanford, CA 94305, USA.

[§]Author for correspondence (reddien@wi.mit.edu)

 P.W.R., 0000-0002-5569-333X

animals, which led to hypotheses about gradients and adult tissue polarity (Morgan, 1898, 1904, 1905). Recent molecular advances have yielded insights into processes underlying the head-versus-tail regeneration choice, but the nature of the core mechanism remains unresolved.

A variety of planarian genes can be inhibited to result in abnormal patterning in regeneration or during tissue maintenance. For example, *β-catenin-1* inhibition causes the regeneration of heads in place of tails and the emergence of heads around the periphery of uninjured animals during tissue turnover (Gurley et al., 2008; Iglesias et al., 2008; Petersen and Reddien, 2008). Study of body plan alteration phenotypes led to the discovery of position control genes (PCGs) (Reddien, 2011; Witchley et al., 2013). PCGs display constitutive regional expression and either a patterning-abnormal RNAi phenotype or are predicted to be part of a planarian patterning pathway. PCGs are notable in constituting adult positional information and are predominantly expressed by planarian muscle (Witchley et al., 2013). The anterior-posterior (AP) axis is regulated by Wnt signaling, which promotes AP-identity regeneration, AP-pattern maintenance and brain scaling (Kobayashi et al., 2007; Gurley et al., 2008; Iglesias et al., 2008; Petersen and Reddien, 2008, 2009a, 2011; Adell et al., 2009; Sureda-Gómez et al., 2015; Lander and Petersen, 2016; Scimone et al., 2016; Stückemann et al., 2017; Schad and Petersen, 2020; Bonar et al., 2022). After amputation, PCG expression domains become rapidly reset. For example, within 30-48 h of injury, a tail fragment regenerates missing anterior PCG expression domains and re-scales posterior PCG expression posteriorly (Petersen and Reddien, 2008, 2009b; Gurley et al., 2010; Witchley et al., 2013; Wurtzel et al., 2015). The re-scaling of PCGs, along with self-organization of tissues and their progenitors, allows for coherent tissue patterning during regeneration (Atabay et al., 2018; Hill and Petersen, 2018). PCG expression pattern regeneration involves changes in gene expression in pre-existing muscle and expression in new muscle cells made at wounds (Petersen and Reddien, 2009b; Gurley et al., 2010; Reuter et al., 2015; Scimone et al., 2016; Tewari et al., 2019). Planarian regeneration requires this re-setting of positional information (Adell et al., 2009; Petersen and Reddien, 2009b; Hayashi et al., 2011; Gaviño et al., 2013; Owlarn et al., 2017; Scimone et al., 2017, 2022; Tewari et al., 2018).

Different mechanisms could be considered for how missing positional information could return after injury, including the intrinsic properties of reaction-diffusion morphogen systems (Meinhardt, 1982). However, it was found that wound signaling is required for the re-setting of planarian positional information (Petersen and Reddien, 2009a). Wounds generically activate *wnt1* expression (Petersen and Reddien, 2009a; Gurley et al., 2010) in muscle (Witchley et al., 2013); wound-induced *wnt1* is required for posterior PCG activation (Petersen and Reddien, 2009a). *notum,* by contrast, is activated generically by wounding, but only strongly at anterior-facing wounds, where it is required for head regeneration

(Petersen and Reddien, 2011). *notum* encodes a secreted Wnt inhibitory protein (Gerlitz and Basler, 2002; Giráldez et al., 2002; Kakugawa et al., 2015). *wnt1* and *notum* activation occurs within ~6 h of injury and these genes are proposed to form a wound-induced switch: *wnt1* generates a Wnt-high environment at posterior-facing wounds, promoting tail regeneration; *notum* generates a Wnt-low environment at anterior-facing wounds, promoting head regeneration. *notum* is the only known wound-induced gene to display a polarized bias towards anterior-facing wounds and is the earliest known asymmetric transcriptional event at head-versus-tail facing wounds (Wurtzel et al., 2015). *notum* expression asymmetry is therefore a wound-induced readout of polarity information.

A small number of genes, including important regeneration regulatory genes, are generically wound induced selectively in planarian muscle, including *wnt1*, *notum, inhibin, glypican-1, follistatin, wntless* and *nlg-1* (Wenemoser et al., 2012; Witchley et al., 2013; Wurtzel et al., 2015). The planarian musculature includes longitudinal, circular and diagonal body wall muscle fibers, as well as dorsal-ventral and intestinal muscle (Hyman, 1951), each with unique molecular identities (Scimone et al., 2017, 2018). Among the select group of muscle wound-induced genes, *notum* and *follistatin* are unique in being specifically wound induced in longitudinal muscle (Scimone et al., 2017). Longitudinal muscle fibers are oriented along the head-to-tail axis, and are composed of numerous individual mononucleated muscle cells (Gelei, 1927; Hyman, 1951). This raises the possibility that polarity information is harbored locally, within longitudinal muscle cells, which themselves would be asymmetrically injured at anterior-versus-posterior-facing wounds. How *notum* is selectively activated in longitudinal fibers at anterior-facing wounds is therefore a central question to address for understanding planarian regeneration.

Previous work indicates that Activin signaling and a non-canonical Wnt11 and Dishevelled pathway are required for asymmetric *notum* activation at wound sites (Cloutier et al., 2021; Gittin and Petersen, 2022). The impacts of perturbing these pathways depends on muscle turnover for the disruption of wound-induced *notum* polarity. This indicates the importance of these pathways in acting prior to injury for establishing a polarity that can be read-out after injury; however, how muscle cells convert existing polarity information into *notum* expression asymmetry upon wounding is not known. In this study, we aimed to identify the molecular properties necessary for asymmetric activation of *notum* after wounding in existing longitudinal muscle. Using single-cell RNA sequencing (scRNA-seq), gene function perturbation and cytoskeletal inhibition experiments, we identified roles for the gene *egal-1* and microtubules in the selective activation of *notum* at anterior-facing wounds.

## RESULTS
### Single-cell RNA sequencing defines the transcriptome of longitudinal muscle cells expressing wound-induced genes
We reasoned that some components mediating polarized wound-induced *notum* expression might also be wound induced in longitudinal muscle cells. To identify such genes, we performed single-cell RNA sequencing (scRNA-seq) on wounded and control animals. Prior planarian scRNA-seq experiments enriched for muscle progenitors, but were sparse for mature and wounded longitudinal muscle cells (e.g. Fincher et al., 2018; Scimone et al., 2018). We therefore developed a gentler cell isolation procedure that did not use FACS, reasoning that wounded muscle cells might be filtered or lost during FACS (see Materials and Methods).

We isolated tissues at anterior- and posterior-facing wounds from pre-pharyngeal regions 18 h post-amputation (hpa). Amputation

planes were selected such that the same cell region was utilized for both wound-face orientations and for an uninjured control sample (Fig. 1A). scRNA-seq was performed using 10X and cell identities in clusters were annotated with known tissue-specific markers (Fig. 1B, Fig. S1). Muscle-specific markers, *collagen* and *tropomyosin*, were used to identify a muscle cluster (Fig. 1B). We subclustered the muscle cell data and identified cells that expressed *myoD* and *snail*, which encode transcription factors specifically expressed in longitudinal muscle (Scimone et al., 2017). The prominent longitudinal muscle subcluster was then isolated and subjected to a further round of subclustering, resulting in five longitudinal muscle subclusters. One such subcluster comprised only cells from injured animals, but no cells from the control, 0 hpa tissue (Fig. 1C). Known muscle wound-induced genes displayed enriched expression in this cluster, including *notum* (ranked 28 among cluster-enriched genes ordered by $\log_2$ fold change), *fst* (ranked 6), *inhibin* (ranked 30), *nlg-1* (ranked 4) and *wntless* (ranked 17) (Fig. 1D). Longitudinal muscle cells were the only muscle type that displayed a distinct subcluster of cells expressing wound-induced genes (Fig. S2A).

The longitudinal muscle subcluster displaying wound-induced gene expression was utilized to identify candidate wound-induced genes in muscle using differential gene expression analysis in Seurat. Seventy-one genes displayed significantly enriched expression in this cluster compared to other longitudinal muscle subclusters ($P$adj<0.05, $\log_2$ fold change>1) and were selected as candidate wound-induced genes in longitudinal muscle cells. We assessed the expression specificity of these genes by determining the $\log_2$ fold increase in their expression across different tissue and muscle cell types (Fig. 1E, Fig. S2B,C). The expression of the top candidate wound-induced genes in longitudinal muscle was robustly expressed in wounded longitudinal muscle and either not expressed or expressed at lower levels in other muscle clusters (Fig. 1E,F). *notum* was asymmetrically induced in the longitudinal muscle cells from anterior-facing compared to posterior-facing wounds, supporting the ability of these data to identify known features of wound-induced genes (Fig. 1E). Furthermore, the previously known canonical muscle wound-induced genes (*notum*, *fst*, *inhibin*, *nlg-1*, *glypican-1*, *wnt1* and *wntless*) also displayed upregulated expression in muscle after wounding (Fig. 1E, Table S1). Most identified muscle wound-induced genes displayed more robust expression in longitudinal versus other muscle types (Fig. 1E); this could reflect the fact that transverse wounds injure a higher fraction of longitudinal muscle than circular muscle (Scimone et al., 2017) or possibly that longitudinal muscle cells are more competent to display a wound-induced transcriptional response than other muscle cell types.

### Identification of wound-induced gene expression dependent on longitudinal muscle with bulk RNA sequencing
Longitudinal muscle cells are maintained in continual cellular turnover, involving cell production from neoblast stem cells (Reddien, 2018). Neoblasts express fate-specific transcription factor genes (FSTFs) that can promote production of tissue-specific progenitors (Reddien, 2022). RNAi of FSTFs can cause continual depletion of a target tissue because of failed cell production during turnover. We analyzed RNA-sequencing data from wound sites of *myoD* RNAi tail fragments from Scimone et al. (2017), which had reduced longitudinal muscle cell number, at 0, 6 and 24 hpa. The goal of this approach was to apply a second method to identify wound-induced genes that are dependent on longitudinal muscle cells for their expression (Fig. 2A). We sought genes that were upregulated in control animals between 0 h and either of the post-amputation

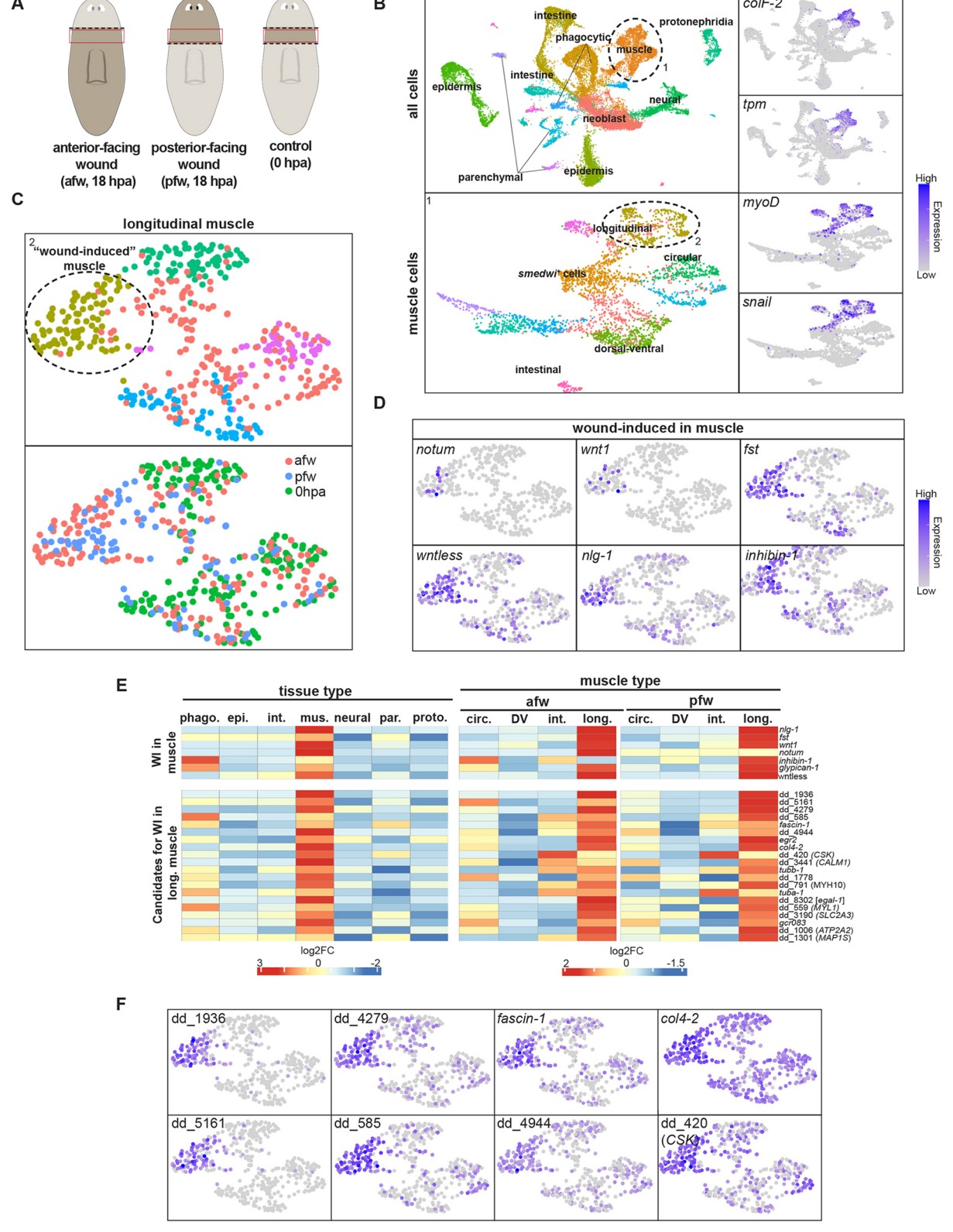

**Fig. 1. scRNA-seq reveals a cluster of longitudinal muscle cells expressing wound-induced genes.** (A) The 10x scRNA-seq experimental workflow. The pre-pharyngeal region taken for sequencing is outlined in red. Dotted line indicates amputation plane. (B) UMAP plots of cells from regions sequenced in A. (C) UMAP plots showing the subclusters of the longitudinal muscle cells. Cells colored by source (afw, anterior-facing wound; pfw, posterior-facing wound; or 0 hpa) show no 0 hpa cells in the circled subcluster. This subcluster was annotated as 'wound-induced' muscle. (D) High expression of muscle wound-induced (WI) genes in the cluster annotated as 'wound-induced' muscle from C. (E) Heatmap of log₂ fold change (log₂ fold change) in gene expression comparing uninjured control (0 hpa) and 18 hpa conditions. Known wound-induced genes and the top 20 additional candidates are shown (red, high log₂ fold change; blue, low log₂ fold change). Expression values were incremented by 1 prior to Z-score normalization. (F) Expression of the top eight wound-induced genes in a wound-induced longitudinal muscle cluster.

timepoints, but that were upregulated to a lesser extent in the *myoD* RNAi dataset. We analyzed the top 200 genes that were significantly wound induced at both 6 hpa and at 24 hpa. We assessed the expression of these genes in the *myoD* RNAi dataset. Among the 200 genes upregulated at each time point, 39 genes had a $\log_2$ fold change of >0.1, comparing *myoD* RNAi to control data at 6 hpa, and 71 genes had a $\log_2$ fold change of >0.25, comparing *myoD* RNAi to control data at 24 hpa (Fig. 2B, Fig. S3, Table S2). Although not all of the expression of the genes within this fold-change cutoff was statistically significant, we reasoned that some of the genes might still be biologically relevant because *myoD* RNAi incompletely eliminates longitudinal fibers and because some genes might have expression in other locations that prevented the changes from being significant with the power of this experiment. We selected 92 of these genes for further analysis, as detailed below.

### Assessment of wound-induced genes in longitudinal muscle fibers with fluorescent *in situ* hybridization

We assessed the expression of 139 genes from the scRNA-seq and *myoD* RNAi bulk RNA sequencing data *in vivo* with fluorescent *in situ* hybridization (FISH) at 18 hpa. 14 and 13 genes were shared between the scRNA-seq datasets and the 6- and the 24-h bulk RNA-seq, respectively. Longitudinal muscle cells were labeled with pooled RNA probes for *myoD* and *snail*. Eighteen genes displayed detectable wound-induced expression in muscle by FISH (Fig. 2C, Fig. S4A). For the remaining genes, some might be wound induced in reality but have transcript levels too low for clear detection by FISH, others were expressed broadly, potentially obscuring wound-induced signal, and some might have been false positives in the sequencing datasets. We assessed the specificity in expression of these 18 genes at wound sites for longitudinal muscle in the FISH and scRNA-seq data (Fig. 2D,E, Fig. S4A-C). Expression of three of the 18 genes, dd_1936, dd_8302 and dd_5161, showed high specificity to longitudinal muscle, with >90% of the cells expressing these genes being longitudinal muscle cells (Fig. 2D). dd_8302 was previously noted to be a gene expressed in muscle, and wound induced in bulk planarian RNA sequencing and microarray data (Wenemoser et al., 2012; Wurtzel et al., 2015). There was also an enrichment of wound-induced *wnt1* expression in longitudinal muscle (Fig. S4D). Among the top-ranked genes in the scRNA-seq dataset that showed highly enriched expression in longitudinal muscle by FISH, dd_1936 (ranked 1) and dd_5161 (ranked 2) were also expressed in other cell types in uninjured conditions (Fig. S4E,F). These two genes encode short peptide sequences of 87 and 79 amino acids, respectively, and have no clear BLAST hits in other organisms. dd_8302 was analyzed in further detail, as described below.

Among the 18 genes with wound-induced expression in longitudinal muscle fibers was the early growth response transcription factor-encoding *egr2* gene (ranked 9; 84.9% of expression in longitudinal fibers). Egr genes act as immediate wound response genes in planarians and across metazoans (Wenemoser et al., 2012). An Egr gene promotes regeneration in the acoel *Hofstenia miamia* (Gehrke et al., 2019). Additional wound-induced genes included: dd_4028 (ranked 15), which encodes a protein similar to human ferric chelate reductase 1 (FRSS1), with roles in inhibition of ferroptosis (Liang et al., 2024); and dd_8461, which encodes a protein similar to the dual-specificity protein kinase Clk2. Clks phosphorylate key proteins in various signaling pathways (Song et al., 2023). *lrfn-1* (ranked 21) encodes a transmembrane glycoprotein involved in neural development, synapse formation, microbial recognition and immunity (Konakahara et al., 2011). *fascin-1* (ranked 7) had moderate expression specificity to longitudinal muscle at wounds

(77.1%). Fascin is an actin-bundling protein involved in regulation of cytoskeletal dynamics (Otto et al., 1979; Adams, 2004). Six genes encoding proteins without clear homology were also identified as wound induced in muscle.

Calcium levels can serve as a cue in some contexts to direct wound healing and regeneration. Genes encoding candidate calcium-binding or calcium-regulated proteins were present in this 18 gene set, including *calreticulin-1* (ranked 47), dd_3441 (*CALM1*, *calmodulin1*; ranked 12), dd_4944 (containing a predicted calcium-binding EF-hand domain; ranked 8) and dd_7046 (ranked 50; *MYLK*, *myosin light chain kinase*) (Fig. 2C). CALM1 proteins act as $Ca^{2+}$ sensors to regulate multiple calcium-dependent signaling pathways (Chin and Means, 2000). Calreticulin is an ER-localized calcium-binding protein, primarily involved in calcium storage and homeostasis (Milner et al., 1991). MYLK regulates the cytoskeleton in muscle through a calcium/calmodulin-dependent phosphorylation of myosin light chains (Kamm and Stull, 1985). RNAi inhibition of a gene encoding a $Ca^{2+}$ channel ($Ca_v1B$) in the planarian species *Dugesia japonica* resulted in bipolar head formation during regeneration, indicating a potential role for calcium in regeneration polarity (Nogi et al., 2009; Chan et al., 2017).

The microtubule protein-encoding *tubulin alpha-2* gene (ranked 48), *tubulin alpha-1* (ranked 16) and *tubulin beta-1* (ranked 13) were among the top genes with enriched expression in the wounded longitudinal muscle subcluster (Fig. 2F, Fig. 1E). However, FISH was insufficient to yield a clear assessment of the wound-induced nature of *tuba-1* and *tubb-1*, possibly because of their broad expression (Fig. S4C). dd_3564 (ranked 26), encodes a protein similar to microtubule-associated protein 1B (MAP1B) and was also broadly expressed (Fig. S4B,C). MAP1-family proteins stabilize microtubules, but can also interact with other cellular components (Ciani et al., 2004; Yang et al., 2012). The upregulation of these genes in longitudinal muscle at wound sites raises the possibility that cytoskeletal regulation has important roles in the biology of longitudinal muscle cells at wounds.

### dd_8302 (*egal-1*) is expressed specifically in longitudinal fibers

One gene, dd_8302 (ranked 3), was the focus of our further investigation because of its specific wound-induced expression in longitudinal muscle fibers (Fig. 2D) and its putative homology to proteins involved in microtubule-dependent transport in other systems (see below). We named this gene *Smed-egalitarian-like-1* (*egal-1*) for reasons detailed below. 95.2% of dd_8302+ cells at wounds expressed the muscle marker *collagenF-2* (*colF-2*), demonstrating the specificity of this wound-induced expression for muscle (Fig. 3A). 95.6% and 92% of dd_8302+ cells at wounds were positive for *myoD*/*snail* transcripts (probe pool) marking longitudinal muscle at 18 hpa (Fig. 2C) and at 6 hpa (Fig. 3A), respectively. By contrast, only 8.4% of dd_8302+ cells were positive for *nkx-1*, a circular muscle marker (Fig. 3A). 97.3% of *notum*+ cells were dd_8302+, yet only 41.6% of dd_8302+ cells were *notum*+, suggesting that only a subset of longitudinal fibers responding to wounding activate *notum*. In uninjured animals, no specific dd_8302 expression was detected by FISH (Fig. 3B); however, in scRNA-seq data, a low level of dd_8302 expression was apparent in uninjured muscle (Fig. S4C).

We also noted that dd_8302 was induced rapidly in longitudinal muscle, compared to other muscle wound-induced genes (Fig. 3B,C). In RNA-sequencing data from planarian wound sites (Liu et al., 2013; Wurtzel et al., 2015), dd_8302 was robustly detected by 4 hpa, closer to its peak level than other muscle

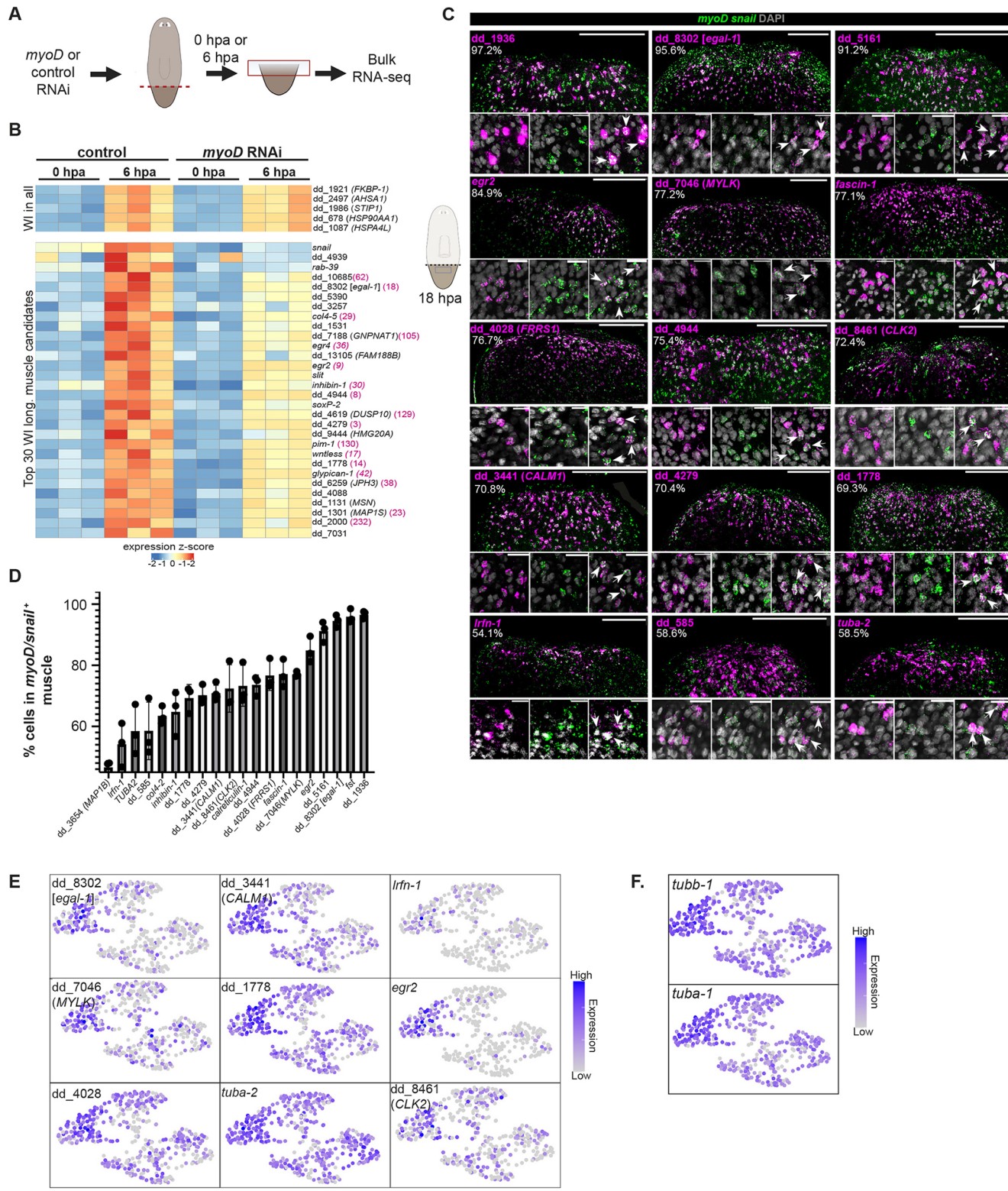

**Fig. 2. Identification of wound-induced genes with varying specificity to longitudinal muscle.** (A) Schematic of experimental workflow (using data from Scimone et al., 2017). (B) Heatmap of genes upregulated at 6 hpa compared to 0 hpa control animals. Top 30 genes downregulated in injured *myoD* RNAi (ranked by log$_2$ fold change) and five genes that are wound induced in all cells (Wurtzel et al., 2015). Each column is a biological replicate. Numbers in brackets indicate the ranking of the enrichment of these genes in the wound-induced longitudinal muscle cluster from the scRNA-seq data. (C) Co-expression of *myoD*/*snail* (probes pooled) with muscle wound-induced genes at 18 hpa. For each gene, the percentage of wound-induced gene-positive cells that also co-express *myoD*/*snail* was calculated and averaged from three animals in one experiment. Scale bars: 200 µm (top); 20 µm (bottom). (D) Bar graph showing the percentage of cells at the wound site that express a wound-induced gene and also colocalize with longitudinal muscle markers. Data are mean±s.d., *n*=3. (E,F) Expression level of the selected genes (E) and tubulins (F) shows higher expression in the wound-induced subcluster of longitudinal muscle than in the non-wound-induced longitudinal muscle subclusters.

wound-induced genes at this time point (Fig. 3C, Fig. S5). With FISH, robust, widespread wound-induced expression of dd_8302 was detected by 3 hpa, whereas essentially no wound-induced

notum expression was detected at this timepoint (Fig. 3B). The strong and early activation of dd_8302 at wounds, preceding notum activation, and the specificity of dd_8302 activation to longitudinal

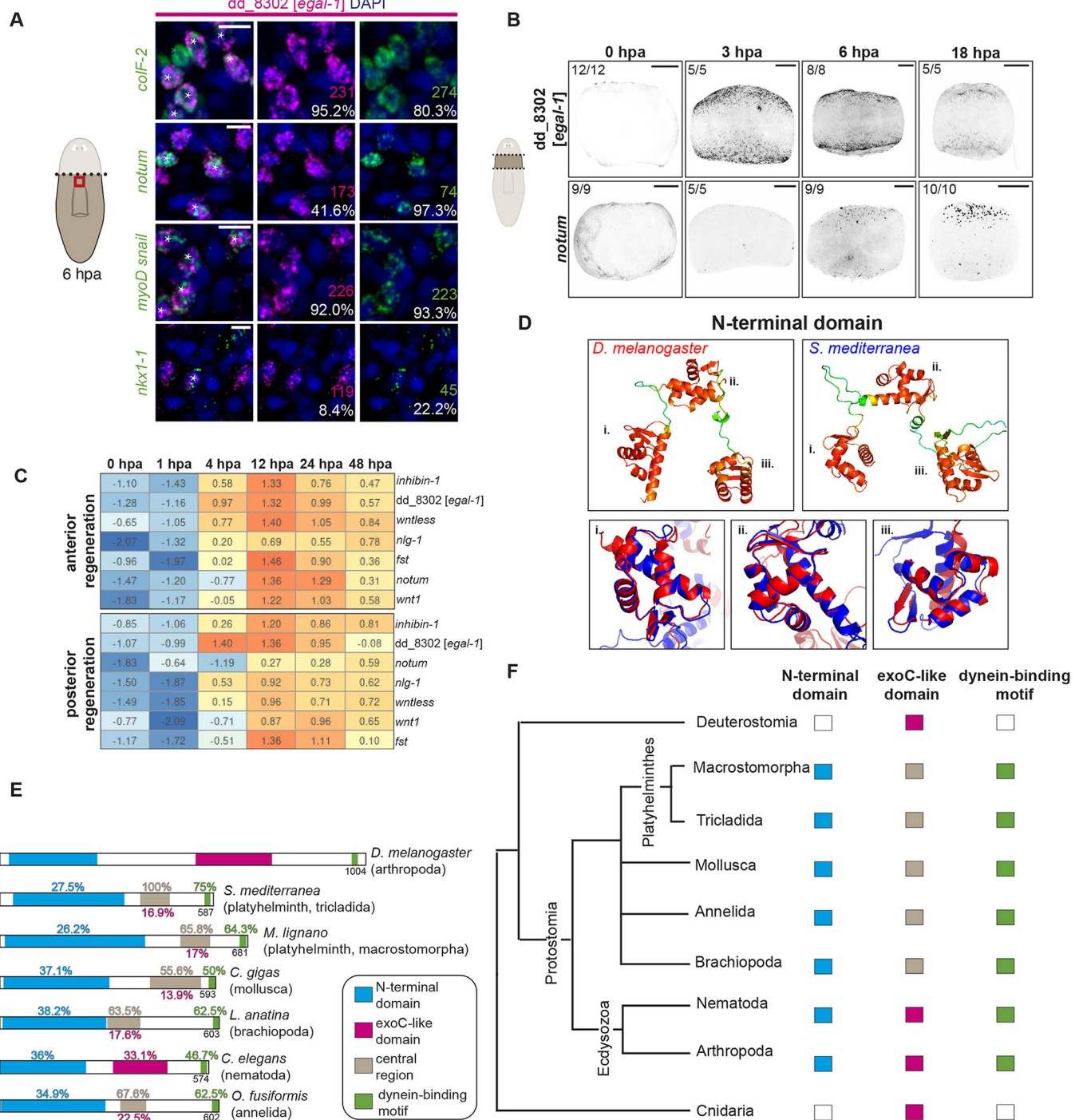

Fig. 3. egal-1 is an early wound-induced gene expressed specifically in longitudinal muscle fibers. (A) Co-expression of egal-1 (dd_8302) with colF-2, notum, myoD and snail pool (longitudinal muscle markers), or nkx1-1 (circular muscle marker) at 6 hpa. White stars indicate double positives. Percentage indicates the fraction of double-positive cells out of the number of single-positive cells shown above, with each number colored to match the corresponding gene. Scale bars: 10 μm. (B) Expression of egal-1 and notum at wound sites following amputation. egal-1 expression is detected at wounds as early as 3 hpa. Scale bars: 200 μm. (C) Heatmap of bulk RNA sequencing data from Wurtzel et al. (2015). Numbers are representative of $\log_2$ RPKM values. (D) AlphaFold2 prediction of a previously unreported N-terminal domain in Egalitarian of D. melanogaster (14-256 bp) and in S. mediterranea Egalitarian-like-1 (37-342 bp). Confidence levels of prediction: blue is low (pLDDT<50), green/yellow is medium (pLDDT 50-90) and red is high (pLDDT>90). Alignment of each ordered region was performed independently of the rest of the protein using the align function of PyMOL. (E,F) Evolutionary conservation of the Egalitarian N-terminal domain, a central region within spiralians and a C-terminal motif that was characterized as dynein binding in Drosophila. Percentage identities to Drosophila domains or to the planarian central region (gray) are shown.

muscle made it a good candidate for a role in wound-mediated processes involved in regenerative patterning.

### dd_8302 (egal-1) encodes an Egalitarian-like protein

dd_8302 encodes a protein that has regions of similarity to proteins found in other organisms. We identified a previously unreported predicted domain in the N-terminus of the dd_8302 protein that displays similarity to an N-terminal region in proteins from other phyla, including arthropods, nematodes, annelids and molluscs, but not clearly in deuterostomes (Fig. 3D,F). Two other *S. mediterranea* genes also encode similar proteins to dd_8302: dd_6723 and dd_7434. Neither dd_6723 nor dd_7434 were wound induced, nor were they specifically expressed in longitudinal muscle cells; we therefore did not study them further (Fig. S6). The N-terminal candidate conserved domain in the predicted *S. mediterranea* dd_8302 protein is 312 amino acids, and its percentage identity to proteins across these phyla ranged from 27.4% to 45.7% (Fig. 3D, Fig. S7). Whereas most of the proteins with similarity to this N-terminal region encode proteins of unknown function, in *Drosophila* the most similar protein to dd_8302 is Egalitarian. *egalitarian* mutant *Drosophila* fail to establish oocyte polarity, and mutants with disruptions in an Egalitarian dynein-binding motif, oocyte specification may occur; however, oocyte polarization is not maintained (Mach and Lehmann, 1997; Navarro et al., 2004). *Drosophila* Egalitarian has RNA-binding activity and is required for the asymmetric localization of developmentally important transcripts, such as *oskar*, *bicoid* and *gurken* in oocytes (Mach and Lehmann, 1997; Bullock and Ish-Horowicz, 2001; Sanghavi et al., 2016). We used Alphafold v2 to assess the predicted structure of this previously unreported candidate N-terminal domain. The predicted N-terminal domain structure of dd_8302 and *Drosophila* Egalitarian were similar and comprise three similar well-folded modules, or ordered regions, connected with less structured linkers (Fig. 3D, Fig. S8). The predicted folded modules in *Drosophila* Egalitarian also aligned to the predicted folded modules from *S. mediterranea* (Fig. 3D). The structure of this N-terminal domain was conserved across Spiralian clades with high structural similarity (Fig. 3E, Fig. S8). The very N-terminus of *Drosophila* Egalitarian can interact with the protein Bicaudal-D (Mach and Lehmann, 1997; Dienstbier et al., 2009), with this interaction region overlapping the region of conserved N-terminal sequence with planarian dd_8302.

*Drosophila* Egalitarian contains a central 3′-5′ exonuclease-like domain that is needed for RNA binding. Whereas mutating critical catalytic amino acids in this exonuclease domain did not interfere with *in vivo* function or capacity to bind RNA, deletion of this 164 amino acid region disrupted Egalitarian function without destabilizing the protein (Navarro et al., 2004; Dienstbier et al., 2009). How other aspects of this domain contribute to RNA binding is unknown. The level of sequence identity in this exonuclease-like domain is low in dd_8302 and its planarian paralogs (16.9% for dd_8302 compared to *Drosophila* Egalitarian, Figs S7, S8) and there is not a clear predicted exonuclease active site in the planarian protein. Therefore, whether the low level of sequence similarity in this region reflects common ancestry of a protein domain between the *Drosophila* and planarian proteins cannot be clearly determined. This central region in the planarian predicted proteins does, however, also align to a central region in proteins from the mollusk, annelid and brachiopod phyla, with >50% identity in all cases (all in the Spiralia superphylum) for proteins that also contained the conserved N-terminal domain. The predicted AlphaFold2 structure of this central region that ranges from 67 to 71 amino acids long also showed clear structural similarity across

species (Fig. S8). This suggests that a conserved central region exists, at least in these Spiralian clades.

The C-terminal end of *Drosophila* Egalitarian can interact with dynein light chain (Navarro et al., 2004). This C-terminal motif displays a high degree of similarity (75% identity) to a C-terminal motif in the predicted dd_8302 protein (Fig. 3E, Fig. S7). This same motif is also found in the two planarian dd_8302 paralogs, and in the proteins containing the conserved N-terminal domain with dd_8302 from *C. elegans,* molluscs, annelids and brachiopods. Overall, these similarities, at least for the N-terminal domain and the C-terminal motif, suggest that the dd_8302 and *Drosophila egalitarian* genes are derived from a common ancestral gene. However, the degree of similarity that exists in the biochemical function(s) of *Drosophila* Egalitarian and dd_8302 is unclear and will be an important target for future molecular work. We therefore named the dd_8302 gene *egal-1* (*egalitarian-like-1*) to recognize the predicted common evolutionary origin of this gene and *Drosophila egalitarian*, but used the added '-*like*' to highlight the fact that there is substantial difference in the central protein region in these species (where the *Drosophila* Exonuclease-like domain resides).

### Egal-1 protein localizes throughout muscle fibers at wounds

We made an antibody to Egal-1 by synthesizing a recombinant 6X-His-tagged Smed-Egal-1 N-terminal fragment in *E. coli* and immunizing rabbits. The resultant polyclonal antibody showed little signal in freshly amputated animals, but showed signal in cells extending into a fibril AP-oriented pattern at wound sites (Fig. 4A,B). Observed signal was ablated by RNAi of *egal-1* but not by RNAi of the *egal-1* paralogs (*egal-2*, dd_6723 and *egal-3*, dd_7434), indicating that the antibody is specific to Egalitarian-like-1 (Fig. 4B, Fig. S9). Egal-1 signal in fibers connected to cytoplasm around nuclei that had *colF-2*-transcripts, a marker of muscle at both anterior- and posterior-facing wounds, demonstrating muscle specificity (Fig. 4C). Egal-1 was broadly distributed throughout longitudinal muscle cell fibers and some concentration was also apparent in foci near muscle nuclei (Fig. 4C). FISH combined with immunolabeling showed that Egal-1 colocalized with wound-induced *notum* expression at anterior-facing wounds (Fig. 4D). Wound-induced Egal-1 signal colocalized with the muscle fiber antibody 6G10, showing Egal-1 protein distributed along longitudinal muscle fibers; this includes muscle cells with a fiber ending at the wound, consistent with signal existing in wounded longitudinal muscle cells. Egal-1 protein in muscle fibers, as well as the Egal-1$^+$ perinuclear focus, showed some colocalization with fibril α-tubulin immunostaining (Fig. 4E, Fig. S9B,C). The presence of α-tubulin within Egal-1$^+$ longitudinal muscle cells indicates that microtubules are present in these fibers.

### egal-1 is required for polarized expression of notum at wounds

We inhibited *egal-1* with RNAi: *egal-1* was required for asymmetric expression of *notum* at wounds (Fig. 5A). Specifically, *notum* was expressed at anterior-facing wounds as normal, but was now also expressed at posterior-facing wounds, and in longitudinal muscle cells (Fig. 5B). The phenotype was incompletely penetrant, with 3/15 animals displaying high levels and 10/15 animals displaying moderate levels of *notum* at posterior-facing wounds (Fig. 5A). Quantifying the number of *notum*$^+$ cells at both wound types across *egal-1* RNAi animals showed a significant difference in the ratio of *notum*$^+$ cells at posterior- to anterior-facing wounds compared to the control, with no significant difference in absolute *notum*$^+$ cell number at anterior-facing wounds (Fig. 5C, Fig. S10A,B). This phenotype was notable, given the sparsity of genes known to affect polarized

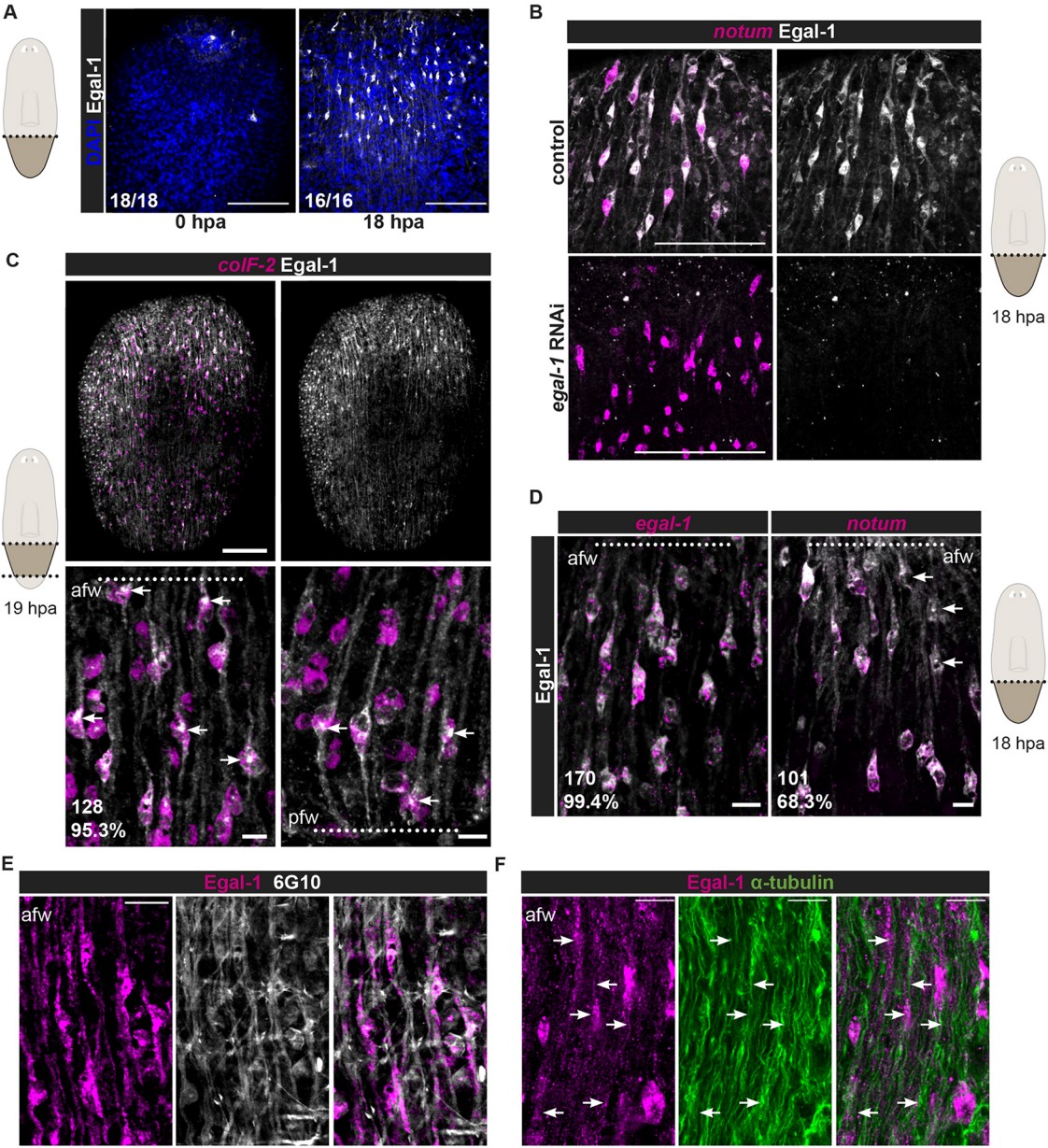

**Fig. 4. Egal-1 protein is expressed in planarian muscle fibers after wounding.** (A) Egal-1 protein expression is induced at wound sites. (B) Egal-1 protein is present in cells at wounds that express *notum* transcripts in control animals. Egal-1 localizes in projections extending from the cell body of the muscle cell. Loss of Egal-1 protein in a wounded *egal-1* RNAi animal, confirming Egal-1 antibody specificity. Anterior-facing wound in a tail fragment at 18 hpa is shown. (C) Egal-1 protein is present in muscle cells (*colF-2*+ cells) at anterior- and posterior-facing wounds. Egal-1 protein extending into projections from the muscle cell body are apparent; arrows indicate foci near the nucleus that are Egal-1+. (D) Co-expression of Egal-1 protein with *egal-1* and *notum* transcripts at 19 hpa. Arrows indicate Egal-1+/*notum*− cells. The fraction of Egal-1+ cells that are positive for *egal-1* or *notum* is indicated below each image. (E) Localization of Egal-1 throughout longitudinal muscle fibers. 6G10 marks muscle fibers. (F) Egal-1 colocalizes with α-tubulin in longitudinal muscle fibers. Anterior-facing wound of tail at 18 hpa. Scale bars: 100 μm in A,B,C (top); 20 μm in E; 10 μm in C (bottom), D,F.

wound-induced *notum* expression; as a control, we showed this phenotype was not found following RNAi of other wound-induced genes (Fig. S10C). We also examined the expression of the muscle wound-induced genes *wnt1* and *wntless*, and the neoblast wound-induced gene *runt-1*, and found that general responses to injury were not overtly impacted by *egal-1* RNAi (Fig. S11).

### *egal-1* is required throughout the AP axis for polarized expression of *notum* at wounds

In *egal-1* RNAi animals, *notum* was wound induced by 6 hpa, and there was only a subtle defect (4/24 animals) in the asymmetry of this initial wound response. The penetrance and strength of *notum*

expression at posterior-facing wounds increased with time (by 18 and 24 hpa) (Fig. 5D). *notum* is also asymmetrically wound induced throughout the AP axis and at diverse wound types (Petersen and Reddien, 2011). *egal-1* was required for asymmetric wound-induced *notum* expression at three different AP axis planes, including in the anterior and the posterior (Fig. 5E). *egal-1* RNAi animals regenerated heads and tails normally, indicating that ectopic *notum* expression at posterior-facing wounds was not sufficient to cause head regeneration (Fig. S12A). AP axis regeneration is facilitated by the formation of a *notum*+ anterior pole and a *wnt1*+ posterior pole (Reddien, 2018). The *notum* expressed in the anterior pole is different from the cells expressing wound-induced *notum* at

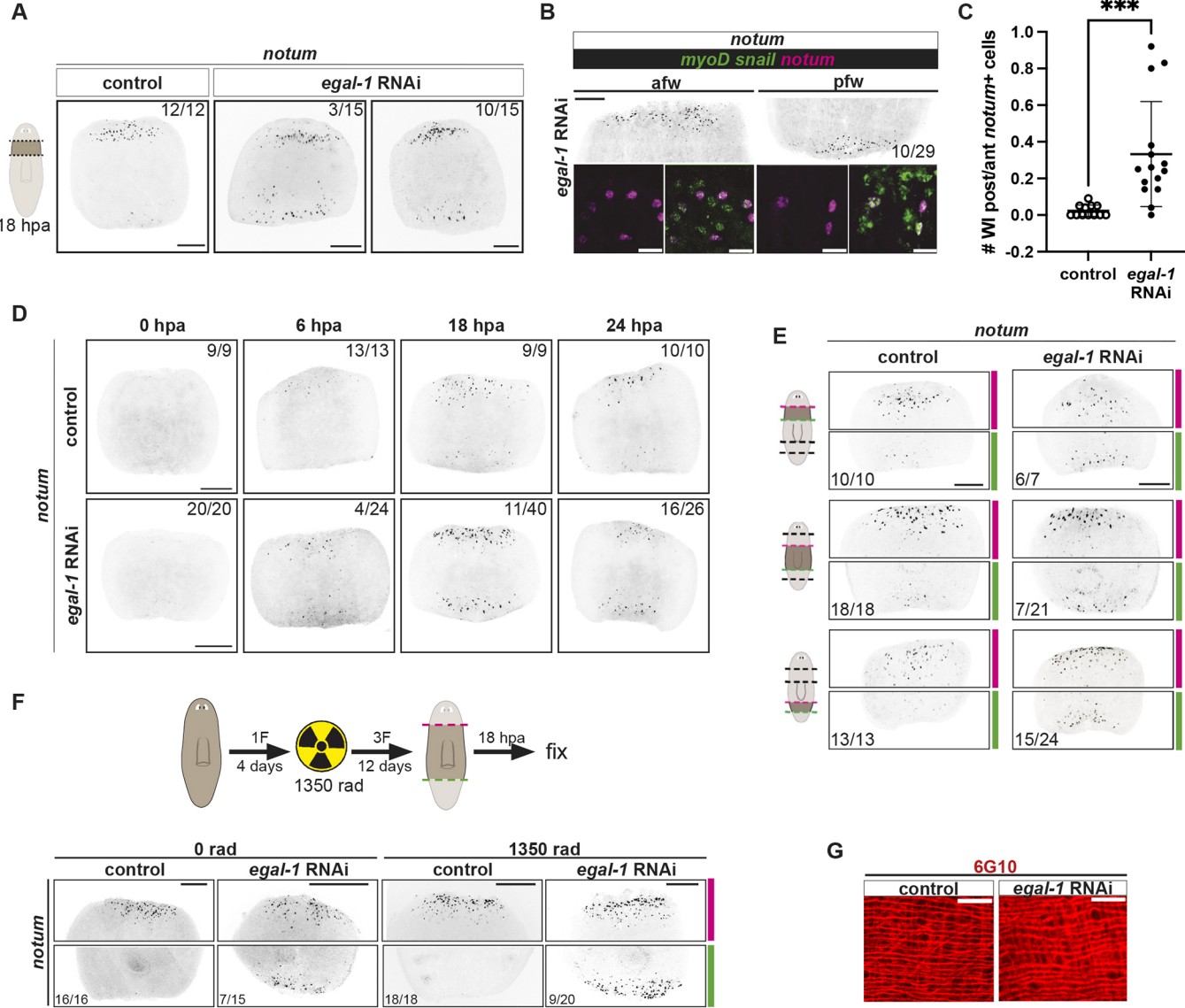

**Fig. 5. *egal-1* RNAi leads to ectopic *notum* expression at posterior-facing wounds.** (A) *egal-1* RNAi animals exhibit ectopic *notum* at posterior-facing wounds with varying strength. (B) Co-expression of *notum* with *myoD*/*snail* probes shows ectopic *notum* expression primarily in longitudinal muscles. (C) Quantification of *notum*⁺ cells in prepharyngeal fragments shown in A. Data are mean±s.d. Control, *n*=12; *egal-1* RNAi, *n*=15 (unpaired *t*-test with Welch's correction, *P*=0.0002). (D) Time course of *notum* expression in *egal-1* RNAi animals. Timecourse of ectopic *notum* expression at posterior-facing wounds. (E) Ectopic *notum* expression is observed along the anterior-posterior axis of *egal-1* RNAi animals. Red (anterior) and green (posterior) dotted lines indicate amputation planes. (F) Top: the experimental workflow. Bottom: FISH shows ectopic *notum* expression at 18 hpa, 16 days after the first *egal-1* RNAi feeding and 12 days after subtotal irradiation. (G) Intact animal muscle fiber organization of *egal-1* RNAi animals showed no visible muscle fiber organization defects. Scale bars: 200 μm in A,B (top),D-F; 20 μm in B (bottom), G.

wounds (Petersen and Reddien, 2011). *egal-1* RNAi animals had no observed defects in asymmetric nucleation of anterior and posterior poles at 3 dpa (Fig. S12B). We also assessed PCG expression, and did not detect defects in the overall pattern of anterior PCG induction at wounds (Fig. S12C). By contrast, *activin-2* RNAi animals and *wnt11-1*, *wnt11-2 RNAi* animals, which also have symmetric wound-induced *notum* expression, did display posterior-facing head regeneration (Cloutier et al., 2021). However, the penetrance of symmetric wound-induced *notum* expression in both those RNAi cases was much greater than was the penetrance of posterior-facing head formation. This similarly with *egal-1* RNAi indicates that *notum* induction at posterior-facing wounds, which can be a molecular marker of abnormal anterior-posterior polarity, can be insufficient for head formation.

## *egal-1* promotes a polarized response to wound orientation in pre-existing longitudinal muscle cells

Wound-induced *notum* is activated in muscle cells present at the time of wounding and is not neoblast dependent (Vásquez-Doorman and Petersen, 2014). Previous studies have shown that *activin-2*, *wnt11-1* and *wnt11-2* are required for muscle cells to generate polarized *notum* activation, and that this defect requires the production of new muscle cells after gene inhibition during cell turnover (Cloutier et al., 2021; Gittin and Petersen, 2022). Therefore, we tested if the *egal-1* RNAi phenotype required newly specified muscle or if *egal-1* played a role in a neoblast-independent process within existing muscle cells for *notum* regulation. Given that neoblasts are the only cycling somatic planarian cells, they can be targeted for depletion using irradiation. We used subtotal irradiation with 1350 rads to reduce neoblasts

(because experiments utilized 2 weeks of *egal-1* RNAi, we chose a dose that would eliminate most neoblasts but allow survival over the course of the experiment). 12 days after irradiation and 16 days after the first dsRNA delivery, *egal-1* RNAi animals displayed ectopic *notum* expression at posterior-facing wounds (Fig. 5F). This is consistent with the possibility that the *egal-1* RNAi phenotype is independent of muscle cell turnover and that *egal-1* functions after injury within pre-existing muscle fibers to regulate asymmetric *notum* expression. Furthermore, we observed no major defects in muscle fiber organization in *egal-1* RNAi animals (Fig. 5G). We suggest that *egal-1* is part of a mechanism that reads out existing polarity information within longitudinal muscle cells after wounding.

### Microtubules are required for planarian regeneration polarity

Among genes that displayed higher expression in longitudinal muscle cells after injury were genes encoding α- and β-tubulin subunits, as described above. This observation, the fact that other candidate cytoskeleton-associated factors were wound induced in our data, and the presence of α-tubulin fibers in muscle cells, led us to address whether microtubules are required for the polarized induction of *notum* at wounds. Microtubules are involved in a large array of polarized cellular processes, and microtubule polymerization and stability can be inhibited pharmocologically. Inhibition of microtubule stability has previously been shown in the planarian species *D. dorotocephala* to cause the regeneration of two-headed animals (McWhinnie, 1955). Therefore, microtubules are an attractive target to investigate for a candidate role in regeneration polarity in the planarian *S. mediterranea.*

Colchicine prevents tubulin dimers from polymerizing, resulting in microtubule destabilization. We utilized 100 μM colchicine applied to planarians for 48 h prior to transverse amputation, after identifying doses and application times that allowed animal survival (Fig. 6A, Fig. S13A, see Materials and Methods). Tubulin is known to be essential for planarian viability (Reddien et al., 2005). Accordingly, much higher doses of colchicine led to animal death. After amputation, pre-pharyngeal fragments were removed from colchicine and examined post-amputation for *notum* expression. In control untreated animals, *notum* was activated at anterior-facing wounds by 6 to 10 hpa, and remained asymmetrically activated at anterior over posterior-facing wounds at 24 hpa. In some colchicine-treated animals, by contrast, *notum* was activated at both anterior- and posterior-facing wounds (Fig. 6B). This defect was incompletely penetrant, with 5/20 animals displaying this defect at 18 hpa and 4/9 displaying symmetric *notum* activation at 24 hpa. The incomplete penetrance raises the possibility that the colchicine level tolerated by animals incompletely perturbed microtubules, with perturbation levels not always sufficient to impede polarized *notum* activation. Alternatively, without microtubules, some process could variably be disrupted such that a polarized response could still sometimes occur. Whereas symmetric *notum* expression was apparent by at least 18 hpa, at 10 hpa the effect was weaker, with only 2/10 animals displaying weak activation of *notum* at posterior-facing wounds. Other muscle wound-induced genes were not overtly impacted by the colchicine treatment applied (Fig. S13D). Animals were fixed at 3 days post-amputation and labeled for the expression of anterior and posterior PCGs. Whereas defects were observed in the morphology of blastemas, no anterior PCG expression was observed in posterior blastema formation, indicating that animals ultimately correctly specified anterior-versus-posterior blastema fates, despite defects in *notum* expression (Fig. S13B,C). These observations raise the possibility that microtubules are required for the correct activation

of *notum* in regeneration polarity, which is significant given the paucity of molecular processes connected to regeneration polarity.

Microtubule perturbation affected polarized *notum* activation across the AP axis (Fig. 6C,D). To further assess the potential connection of microtubules to regeneration polarity, we used two other drugs that interfere with microtubules: nocadazole and taxol. Whereas nocadazole, like colchicine, causes destabilization of microtubules, taxol causes microtubule stabilization. Taxol treatment had no detected effect and nocadazole resulted in robust symmetric expression of *notum* at anterior- and posterior-facing wounds, providing support for the interpretation that microtubule stability is required for asymmetric *notum* activation at wounds (Fig. 6E,F).

Irradiating planarians with 6000 rads depletes all neoblasts within ~24 h. Ectopic *notum* expression was still present in irradiated, colchicine-treated animals (Fig. 6G). Therefore, neoblasts are not acutely required for *notum* expression at posterior-facing wounds in colchicine-treated animals. Additionally, similar to *egal-1* RNAi animals, colchicine-treated animals did not have overtly altered muscle fiber organization (Fig. 6H). Taken together, this suggests that microtubules play a local role to regulate polarity in existing muscle fibers at wounds.

There were comparable levels of *egal-1* transcript expression at both anterior- and posterior-facing wounds in colchicine-treated animals (Fig. S14A). However, colchicine treatment led to lack of signal for the perinuclear foci of Egal-1 protein (Fig. S14B). This suggests that some elements of Egal-1 localization are microtubule dependent.

### DISCUSSION

The mechanisms for activating regeneration programs at wounds is a central problem in regeneration. In salamander limb regeneration, for example, positional memory determines the proximal-distal location of amputation (Otsuki and Tanaka, 2022). Anterior-posterior positional memory is also retained and helps activate regeneration (Otsuki et al., 2025). In planarians, a transverse amputation plane activates head regeneration if anterior facing, and tail regeneration if posterior facing. These features suggest a polarized attribute of the planarian AP axis that enables local interpretation of AP orientation locally at wounds.

Wound signaling acts as an external input to initiate positional information resetting and regeneration in planarians (Petersen and Reddien, 2009b, 2011; Gurley et al., 2010; Owlarn et al., 2017). Although wound signaling is generic, there is a process that must be activated post-injury to yield asymmetric *notum* expression in longitudinal muscle. To explore possible mechanisms underlying this process, we sought wound-induced factors acting within longitudinal muscle fibers to enable polarized activation of *notum* (Fig. 7A). Approximately seven canonical genes are well-known to be wound-induced in planarian muscle within ~6 hpa, two of which are highly selective for longitudinal muscle (Witchley et al., 2013; Wurtzel et al., 2015; Scimone et al., 2017). Several other muscle-expressed genes have also been identified as wound-induced in bulk RNA sequencing (Wurtzel et al., 2015). By utilizing scRNA sequencing without FACS, we describe an additional 18 genes that are activated in muscle at wounds, including three that are highly selective for longitudinal muscle (Fig. 7B). Several of the genes expressed in wounded muscle encode candidate cytoskeletal components, regulators or interacting proteins (Fig. 1E, Table S1).

One wound-induced gene in longitudinal fibers, *Smed-egalitarian-like-1* (*egal-1*), was required for selective *notum* expression – *egal-1* RNAi animals activated *notum* at both anterior- and posterior-facing wounds. *egal-1* transcription was robustly wound induced within

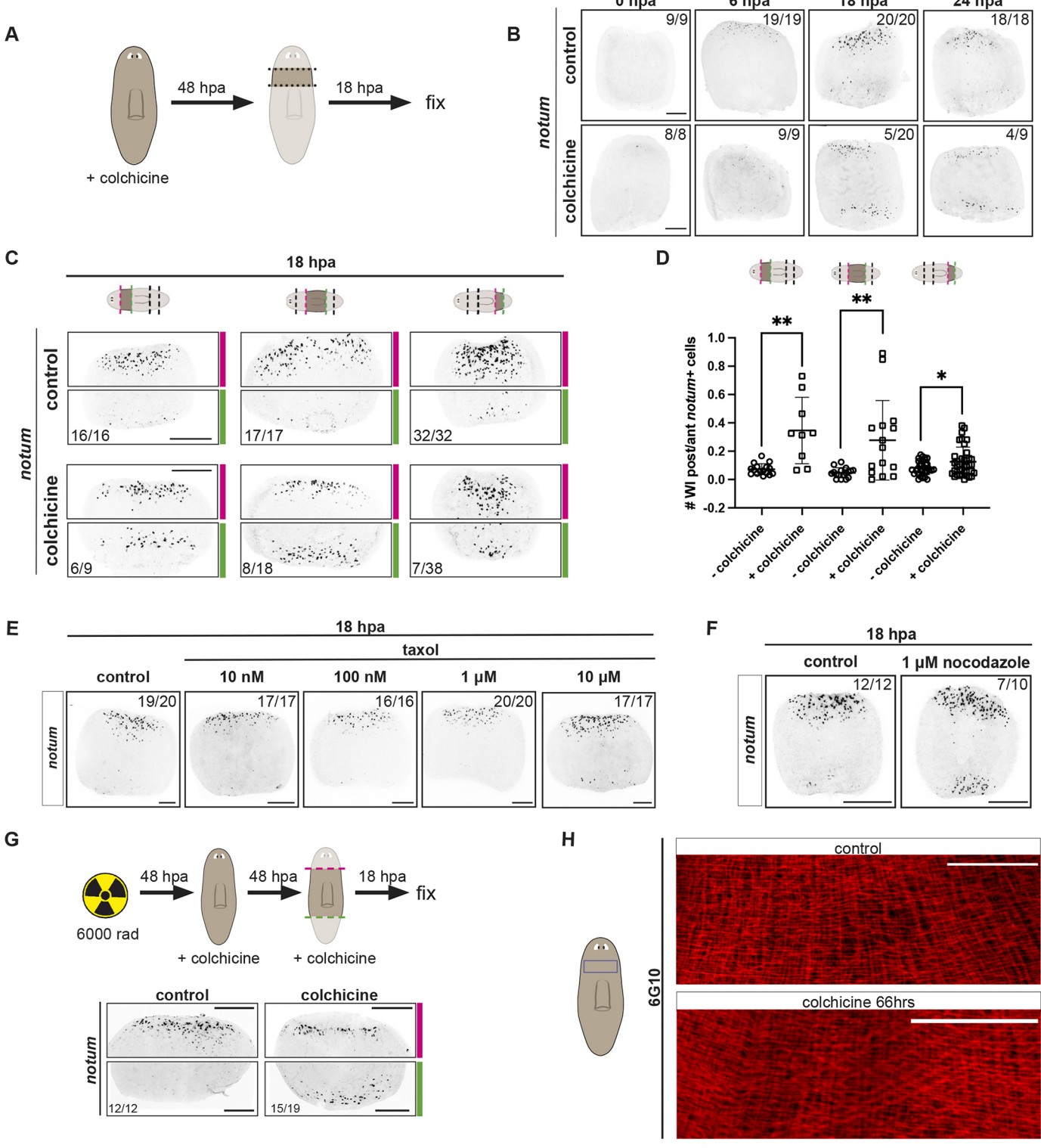

**Fig. 6. Inhibition of microtubule assembly leads to ectopic *notum* expression at posterior-facing wounds.** (A) The experimental workflow. (B) Animals treated with colchicine (100 µM) show ectopic *notum* expression at posterior-facing wounds. (C) Ectopic *notum* expression observed along the AP axis in colchicine-treated animals. Red (anterior) and green (posterior) dotted lines indicate amputation planes. (D) Quantification of *notum*+ cells in fragments, as indicated in schematics above at 18 hpa (unpaired *t*-test with Welch's correction, *P*=0.008, 0.007 and 0.03 respectively from left to right). Data are mean±s.d. (E) Dose-response effect of taxol treatment shows normal *notum* expression at wound sites. (F) Nocodazole-treated animals display ectopic *notum* expression at posterior-facing wounds. (G) Top: the experimental workflow. Bottom: FISH shows symmetric *notum* expression in lethally irradiated (6000 rad) animals treated with colchicine for 48 h pre-amputation and 18 h post-amputation. (H) Intact animal muscle fibers (6G10+) in control and colchicine-treated animals by immunofluorescence. No visible changes in muscle fiber organization was observed. Scale bars: 200 µm in B,C,G; 100 µm in E,F,H.

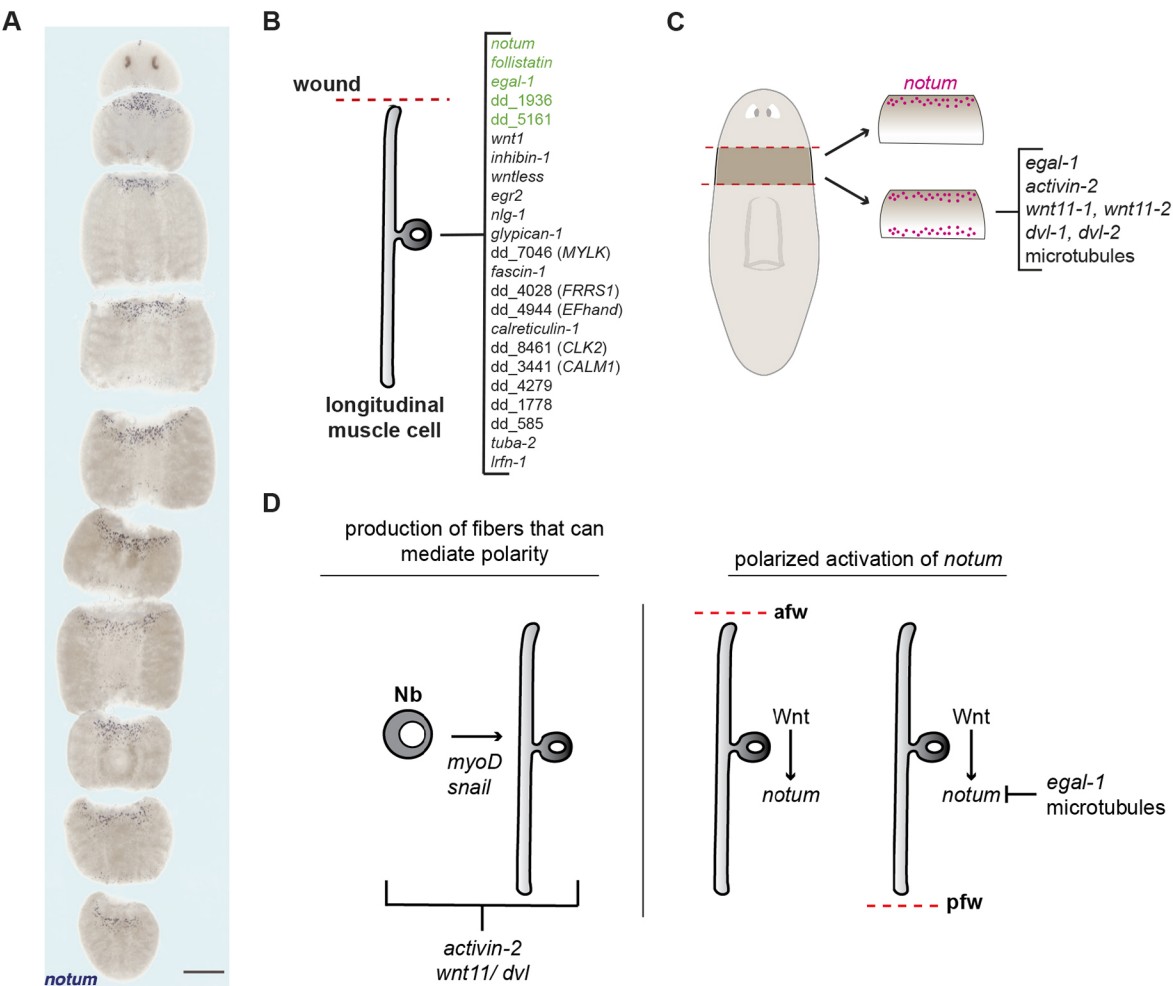

**Fig. 7. Summary of wound response in muscle fibers.** (A) Wound-induced *notum* expression at anterior-facing wounds along the AP axis in a sexual *S. mediterranea.* Scale bar: 500 µm. (B) Wound-induced genes expressed in muscle cells. Genes with high specificity to longitudinal muscle fibers are depicted in green. (C) Normal *notum* expression and ectopic *notum* expression observed when certain factors (*wnt11-1*, *wnt11-2*, *dvl-1*, *dvl-2*, *egal-1*, *activin* and microtubule stability) are perturbed. (D) Model illustrating factors involved in polarized *notum* expression. Wnt11, Dishevelled and Activin pathways require neoblasts to polarize *notum* activation, suggesting a role prior to injury in the formation of muscle that is competent to show a polarized wound response. We suggest that *egal-1* and microtubules are required for a process occurring after injury in longitudinal muscle that prevents wound-induced *notum* expression in longitudinal muscle at posterior-facing wounds.

3 hpa, preceding robust *notum* expression. Egal-1 protein was localized throughout longitudinal muscle cells at wound sites, with some concentration apparent in foci near muscle nuclei. The rapid wound-induced and longitudinal muscle-specific expression of *egal-1* suggests that it may mediate some polarized process at wounds, rather than act to generate polarity prior to injury. *activin-2*, *wnt11* and *dishevelled* are also required for regeneration polarity and appear to act prior to injury to instruct whether the newly made muscle cells will be developed with the capacity to read out wound orientation (Cloutier et al., 2021; Gittin and Petersen, 2022). These findings support a model involving both pre-injury polarization of longitudinal muscle (requiring *activin-2*, *wnt11* and Dvl genes) and processes happening in longitudinal fibers after injury for a polarized transcriptional response (requiring *egal-1*) (Fig. 7C,D).

Despite expression of *notum* at posterior-facing wounds, *egal-1* RNAi animals did not regenerate posterior-facing heads. Several possibilities could explain this. First, despite RNAi robustly removing Egal-1 protein, *egal-1* RNAi animals displayed incomplete penetrance of symmetric *notum* expression, and *notum* was frequently not as robustly expressed at posterior-facing wounds as it was at anterior-facing wounds. Second, although *notum* is required for head

regeneration at anterior-facing wounds, *notum* expression might not be sufficient for head regeneration at posterior-facing wounds – e.g. if there are processes at posterior-facing wounds that override the contribution of *notum* or processes at anterior-facing wounds that collaborate with *notum* for head regeneration that are not ectopically induced by *egal-1* RNAi. These possibilities could be investigated through screens for perturbations that enhance the *egal-1* RNAi phenotype.

Our findings also identify a role for microtubules in regeneration polarity. scRNA-seq data showed that two α-tubulin genes, one β-tubulin gene and one gene encoding a microtubule-associated protein (MAP1B) had increased expression in longitudinal muscle cells at wounds. Microtubules were also present near Egal-1 in longitudinal muscle. These observations and the fact that Egalitarian in *Drosophila* acts with microtubules for asymmetric localization of RNAs led us to explore whether a microtubule-based process contributes to regeneration polarity. Colchicine and nocadazole treatments, which disrupt microtubule polymerization, resulted in *notum* activation at both anterior- and posterior-facing wounds. Colchicine treatment was previously shown to cause posterior-facing head regeneration in the planarian *D. dorotocephala* (McWhinnie,

1955), although we did not observe this outcome in *S. mediterranea* under conditions examined. Colchicine treatment did not drastically alter muscle fiber structure and its impact on *notum* expression at wounds was irradiation insensitive, indicating pre-existing fibers acutely require microtubules for *notum* expression. We suggest that microtubules and Egal-1 act in polarized longitudinal muscle fibers to read out wound orientation and to mediate the correct polarized transcriptional response. Whereas the drug treatments should affect microtubules throughout the body, it is possible that a microtubule-based process mediates polarity in longitudinal muscle cells directly. At anterior-facing wounds, the anterior longitudinal fiber end is removed, whereas the posterior end is removed at posterior-facing wounds. One possibility is that fibers are polarized such that an inhibitor of wound-induced *notum* expression is localized to the anterior side of the fiber, and either cannot be localized or is removed by injury (alternatively, a positive regulator of *notum* could be localized to posterior fiber ends and be removed by injury at posterior-facing wounds). Microtubules and Egalitarian-like-1 could be involved in the transport of some factor(s) (e.g. RNA or protein) involved in this polarized process and this factor could be differentially affected by anterior-versus-posterior fiber end loss. *egal-1* RNAi and microtubule inhibition leads to *notum* activation at posterior-facing wounds, rather than loss at anterior-facing wounds; this favors a model in which an inhibitory process involving Egal-1 and microtubules is active in longitudinal fibers with posterior injury (Fig. 7D). Molecular investigation of any polarized factors and their regulation by Egalitarian will be an important future direction for assessing these possibilities.

Polarity requires a local response in longitudinal fibers at any position across the anterior-posterior axis, with fibers oriented to respond in the same direction. Some process likely sets this body-wide vector, possibly involving Wnt11 (Gittin and Petersen, 2022) or other processes. Planar cell polarity is an example of a process that can generate polarity in cells with the same orientation across large length scales. Some conceptually similar process could in principle impact the coordinated polarity of longitudinal muscle fibers; however, planar cell polarity is not known to affect *notum* transcription but instead is implicated in planarian epidermis cilia polarity (Vu et al., 2019). How longitudinal muscle cells might set their polarity to align to a body-wide vector is thus unknown, but we suggest that microtubules together with Egalitarian-1 act within polarized fibers to mediate a process occurring after injury.

Overall, the identification of microtubules and *egal-1* as regulators of asymmetric wound-induced *notum* expression supports a model in which wound-induced signals translate the possible intracellular polarity of wounded muscle fibers to regulate *notum* expression, ultimately re-establishing body-wide axis polarity. These factors identify major molecular components of the mysterious and fundamental property of planarian regeneration polarity.

## MATERIALS AND METHODS
### Animal husbandry
Experiments were conducted using asexual *Schmidtea mediterranea* (strain CIW4). Animals were maintained in containers containing 1× Montjuïc water (1.0 mmol/l CaCl₂, 0.1 mmol/l MgCl₂, 1.0 mmol/l MgSO₄, 0.1 mmol/l KCl, 1.6 mmol/l NaCl and 1.2 mmol/l NaHCO₃ in Milli-Q water) at 20°C in darkness. They were fed pureed calf liver bimonthly and were starved for 7-14 days being used in experiments.

### 10× single-cell RNA sequencing and analysis
Tissue fragments from the prepharyngeal region of animals amputated 18 h earlier and intact animals were excised using a scalpel, as illustrated in Fig. 1A. Ten large animals were used per condition. The fragments were collected in

Eppendorf tubes containing 1× Montjuïc water, which was then replaced with 0.25% Trypsin-EDTA (1×). The tissues were incubated for 6 min with continuous pipetting for the last 4 min to dissociate the tissue. Following dissociation, the cells were centrifuged at 500 *g* for 5 min then resuspended in calcium-magnesium-free solution (CMFB; 400 mg/l NaH₂PO₄, ,800 mg/l NaCl, 1200 mg/l KCl, 800 mg/l NaHCO₃, 240 mg/l glucose, 15 mM HEPES, pH7.3) with 1% BSA. The solution was filtered using a 40 μm filter. Trypan Blue staining was used to evaluate cell viability and concentration.

The Whitehead Institute Genome Technology Core (WIGTC) processed the cells using the 10X Genomics Chromium Controller and Chromium Single Cell 3′ Library & Gel Bead Kit, following standard protocol. Sequencing was performed using NovaSeqSP 6000 (150×150 paired-end reads) and reads were aligned using a GTF annotation file for Smed_v6 genes (available at https://planmine.mpinat.mpg.de/planmine/model/bulkdata/dd_Smed_v6.pcf.contigs.fasta.zip) within the Smes_g4 genome (https://planmine.mpinat.mpg.de/planmine/model/bulkdata/dd_Smes_g4.fasta.zip). This GTF file was created by mapping all Smed_v6 transcripts to the Smes_g4 genome using BLAT, assigning each transcript to a single genomic location based on the highest alignment score. The transcripts were then consolidated based on their genomic positions before alignment with the Cell Ranger 7.2.0 pipeline.

The Seurat 5.1.0 package was used for 10X data analysis and visualization. The cells were filtered based on the number of UMIs (nCount_RNA) and genes (nFeature_RNA) detected in each cell, and was optimized for each library accordingly. For the anterior-facing wound and posterior-facing wounds, cells with fewer than 2500 and greater than 15,000 UMIs, and fewer than 350 and greater than 2500 genes were removed. For the intact worm, cells with fewer than 4000 or greater than 15,000 UMIs, and fewer than 350 or greater 2500 genes were removed. Cells were normalized using Seurat's NormalizeData function with a scale factor of 10,000, which adjusts the raw expression counts for sequencing depth and scales them to a uniform distribution. Variable features were identified using the FindVariableFeatures function, identifying genes for downstream dimensionality reduction. The data were then scaled using the ScaleData function, and Principal Component Analysis (PCA) was performed on the scaled data using the RunPCA function. Cell clusters were identified using the FindNeighbors function followed by the FindClusters function to group cells into discrete clusters based on their transcriptomic similarity. The identified clusters were visualized using Uniform Manifold Approximation and Projection (UMAP) plots, generated with the RunUMAP function, and further explored with DimPlot to display the spatial arrangement and clustering of cells. UMAP plots of gene expression were created using Seurat's FeaturePlot function. Tissue clusters were annotated using Seurat's FeaturePlot function based on the expression of known tissue-specific genes. Markers of the 'wound-induced' longitudinal muscle cluster (Fig. 1D-F) were identified using the FindMarkers function. A log₂ fold change greater than 1 (>2 fold total difference) and a *P*adj<0.05 threshold was selected because most known muscle wound-induced genes fall within this range (e.g. *wntless*, with an avg_log₂ fold change of 1.1, was the lowest among them; Table S1) and our aim was to capture a broad set of biologically relevant candidates. Heatmaps for tissue enrichment were generated using average expression values obtained using the AverageExpression function and values were z-score normalized by row.

### Fluorescence *in situ* hybridization
All constructs were cloned from planarian cDNA into the pGEM vector (Promega). RNA probes were synthesized as described by Pearson et al. (2009) and whole-mount FISH was performed as previously described (King and Newmark, 2013). Briefly, all fixations for FISH were carried out with 5% NAC treatment for 5 min followed by 20 min 4% formaldehyde fixation. Hybridization was conducted overnight at 57°C and samples were washed the following day. DIG-labeled probes were blocked in 10% Western Blocking Reagent (Roche, 11921673001) PBST solution, while DNP probes were blocked in 5% Western Blocking Reagent and 5% casein. Antibody incubation was performed overnight at 4°C followed by washing then tyramide development. For double or triple FISH experiments, 1% sodium azide was used to inactive the horseradish peroxidase antibody between detection steps.

Fluorescent images were taken using Lieca Stellaris confocal or Lieca SP8 microscope. All images were analyzed using Fiji/ImageJ software. Colocalization of signal was determined by fluorescence intensity around or in DAPI stained nuclei of cells. To quantify double-positive cells for wound-induced gene expression in longitudinal muscle, cells positive for the candidate gene were first identified by viewing the corresponding fluorescence channel alone and determining that the fluorescence signal was around the cytoplasm of a single DAPI$^+$ nucleus. The *myoD* and/or *snail* channel was then overlaid to assess colocalization. All FISH images shown are representative of all images in each condition and are maximal intensity projection, unless showing a specific tissue layer or co-localization of signal in a cell. Live images were taken using a Zeiss Discovery Microscope.

### Structural similarity analysis

Similarity between S. *mediterranea* dd_8302 and *D. melanogaster* Egalitarian was found using BLAST. BLAST of the N-terminal domain to individual phylum databases was conducted to identify proteins with this domain present. MUSCLE was used to perform multiple sequence alignment of best BLAST hits and jalview was used to visualize the data. Protein sequences used to make the alignments and obtain percentage identity scores are provided in Table S3. AlphaFold2 was used to visualize the predicted N-terminal domain structure and sequence alignments were manually adjusted on jalview to mirror predicted structural alignments. We used predicted protein structures to better assess the relationship of candidate homologs across species. Structural similarity scores were obtained using the *Foldseek* easy-search function, which compares AlphaFold2-predicted structures and outputs pairwise structural similarity scores, i.e. bits. These scores were normalized to the maximum bit score converted to distances using the formula $1 - \frac{bits}{\max bits}$. Hierarchical clustering of the pairwise distance values was used to generate a dendrogram representing structural similarity among the proteins.

### Antibody generation from recombinant Egal-1 and immunostaining

Rabbit antisera to Egalitarian-like-1 using peptide or recombinant protein were generated by Genscript; however, antisera from peptides CSKGSRNRPRSAIIN or SSIEADNGHSSIHDC showed no detectible signal in wounded animals fixed with formaldehyde or Carnoy's fixation. Recombinant N-terminal, C-terminal and full-length fragments of Egal-1 were synthesized as *E. coli*-codon optimized geneBlocks (IDT) and cloned as N-terminally 6x-His-tagged constructs with a GSS linker into a pET28 expression vector but only the well-expressed and soluble fragment was the N-terminal construct. To generate purified recombinant Egal-1N for antisera production, LOBSTR BL21(DE3) cells were induced at OD600=0.5 with 0.5 mM IPTG for 3 h at 37°C, lysed with BugBuster plus Lysonase (Novagen) diluted in 600 mM NaCl Ni-NTA buffer [50 mM NaH$_2$PO$_4$ (pH 8.0), 5 mM imidazole, 0.5 mM PMSF and 10% glycerol (v/v)] according to the manufacturer's instructions. Clarified lysate was loaded onto pre-equilibrated Ni$^{2+}$-NTA Hi-Trap agarose (Qiagen; 0.25 ml final bed volume per 1 liter of culture) for 60 min, washed sequentially with four column volumes each of 1 M NaCl, 600 mM NaCl and 400 mM NaCl Ni-NTA buffer, and eluted in multiple fractions with 400 mM NaCl Ni-NTA buffer with 300 mM imidazole (pH 7.5). Protein was concentrated with an Amicon Ultra-4 10 kDa NMWL Centrifugal Filter Unit (EMD Millipore) to 11 mg/ml (Qubit Protein Assay) in protein storage buffer (20 mM HEPES, 500 mM KCl and 20% glycerol at pH 7.8) according to manufacturers' instructions and verified on 16% SDS-PAGE gel stained with Coomassie Blue. All chromatography was performed at 4°C. Buffer was exchanged to PBS for antisera generation in two New Zealand rabbits and purified polyclonal antisera was obtained with an estimated titer of 1:512,000 (Genscript). Immunostaining required an optimized sample fixation of 2% HCl for 30 s at room temperature, 5% NAC in PBS for 5 min under gentle nutation at room temperature with ~1/2 volume of glass vial, fresh ice-cold Carnoy's fixation (6:3:1 ethanol:chloroform:glacial acetic acid) for 2 h on ice, two methanol washes, followed by an overnight bleach of 6% H$_2$O$_2$ in methanol. Animals were rehydrated through 50% methanol/50% PBST (PBS with 0.3% Triton X-100), treated with 2 µg/ml ProteinaseK (Roche) for 10 min and post-fixed for 20 min in 4% formaldehyde in PBST and then were either directly immunostained or first underwent *in situ* hybridization

with DIG-labeled RNA probes as described above. To reduce background, a working aliquot of α-Egal-1N was incubated nutating at room temperature for 2 h and static at 4°C for 3 h with ~2.0 mg acetone powder from uninjured sexual strain S. *mediterranea* and then spun twice for 10 min at 18,242 *g* to remove powder, and stored at 4°C. Acetone powder was generated by adding five volumes of acetone to animals, followed by dounce homogenization on ice in PBS and spinning for 3700 *g* 5′, resuspending in acetone twice before drying overnight on Whatmann filter paper at room temperature. α-Egal-1N was used at 1:100 in 5% inactivated horse serum blocking solution in PBST overnight at 4°C, washed six times in PBST, visualized with tyramide signal amplification after incubation with anti-rb-HRP (Invitrogen, T20924; 1:300) overnight at 4°C, washed six times in PBST with 1:1500 rhodamine tyramide 1:1500 or 1:2000 FITC-tyramide in TSA solution [0.003% H$_2$O$_2$ 20 µg/ml 4IPBA in 100 mM boric acid and 2 M NaCl (pH 8.5)] for 10 min, washed five times in PBST and once in PBS. Animals were mounted in Vectashield and imaged on a Zeiss LSM700 confocal microscope using a 63× NA 1.4 objective with 0.2×0.2×0.57 µm resolution. Linear contrast and LUT adjustments were performed in Fiji with ImageJ 2.0.0. Images are displayed as maximum intensity projections through the muscle layer.

The 6G10 antibody (Ross et al., 2015) produced on formaldehyde-fixed cells from the planarian blastema marks planarian muscle fibers and was used at a 1:100 dilution in 10% inactivated horse serum blocking solution. The anti-tubulin-α Ab-2 (Neomarkers, MS-581-P0) was used at 1:100 in 1% BSA blocking solution. An anti-mouse Alexa Fluor 488- or 568-conjugated secondary antibody (Life Tech, A-10029 and A-11031) was used in a 1:500 dilution.

### Colchicine, nocodazole and taxol treatment

Colchicine (Sigma, C9754-1G), nocodazole (10 µg/ml in DMSO, Sigma, M1404) and Taxol (Tocris, #1097) were prepared in 1× Montjuïc water at the appropriate concentrations. Colchicine was freshly dissolved in 1×Montjuïc water without a vehicle. Nocodazole and Taxol stock solutions in DMSO were diluted to final concentrations, with matching DMSO controls. Optimization experiments tested varying drug concentrations and incubation durations to identify conditions that maximized *notum* expression while maintaining animal survival. Final drug concentrations were as follows: colchicine, 100 µM; nocodazole, 1 µM (with 0.003% DMSO); and Taxol, 10 nM, 100 nM, 1 µM (with 0.1% DMSO) and 10 µM (with 4.3% DMSO). For all drug treatments, animals were incubated in the respective drug solutions for 48 h, with the appropriate DMSO concentration as a control where applicable. After incubation, animals were washed in untreated 1×Montjuïc water, then underwent amputation. Fragments were allowed to recover in untreated 1×Montjuïc water until fixation, unless otherwise stated.

### RNAi

dsRNA was prepared by *in vitro* transcription of PCR products with flanking T7 promoters using T7 reverse transcriptase. The transcription product was precipitated using 100% ethanol at −20°C and annealed after resuspension in RNAse free water. The concentration of dsRNA was variable in each prep with a range of 4-7 µg/ml. dsRNA was mixed with homogenized calf liver and red food dye. Excess food was administered to ensure all animals ate maximally. *egal-1* RNAi animals were fed twice a week for 2 weeks. If animals were still visibly red from the food dye 3 days after the final feeding, they were subject to mild shaking and light exposure to help clear residual gut contents and minimize background signal during FISH. Animals were amputated 4 or 5 days after last feeding.

### Acknowledgements

We thank the Reddien Lab for discussion, and Katherine Malecek for guidance and advice on Egal-1 recombinant protein production for antibody production. We thank the Genome Technology Core for the single cell RNA sequencing.

### Competing interests

The authors declare no competing or financial interests.

### Author contributions

Conceptualization: Y.M., S.M., L.E.C., P.W.R.; Data curation: Y.M., S.M., L.E.C., P.W.R.; Formal analysis: Y.M., S.M., L.E.C., P.W.R.; Funding acquisition: P.W.R.;

Investigation: Y.M., S.M., L.E.C., P.W.R.; Methodology: Y.M., S.M., L.E.C., P.W.R.; Project administration: P.W.R.; Resources: P.W.R.; Supervision: P.W.R.; Validation: Y.M., S.M., L.E.C., P.W.R.; Visualization: Y.M., S.M., L.E.C., P.W.R.; Writing – original draft: Y.M., S.M., P.W.R.; Writing – review & editing: Y.M., S.M., L.E.C., P.W.R.

**Funding**
L.E.C. was a Damon Runyon Fellow supported by the Damon Runyon Cancer Research Foundation (DRG-2428-21) during writing. We thank the Eleanor Schwartz Charitable Foundation for support. P.W.R. is an investigator of the Howard Hughes Medical Institute and an associate member of the Broad Institute. Open Access funding provided by the Howard Hughes Medical Institute. Deposited in PMC for immediate release.

**Data and resource availability**
The scRNA-seq datasets generated and/or analyzed during the current study have been deposited in the NCBI Sequence Read Archive (SRA) under the accession number PRJNA1299472. The accession number of the bulk RNA-seq dataset previously reported is GSE99067 (Scimone et al., 2017). All other relevant data and details of resources can be found within the article and its supplementary information.

**Peer review history**
The peer review history is available online at https://journals.biologists.com/dev/lookup/doi/10.1242/dev.204668.reviewer-comments.pdf

**Special Issue**
This article is part of the Special Issue 'Lifelong Development: the Maintenance, Regeneration and Plasticity of Tissues', edited by Meritxell Huch and Mansi Srivastava. See related articles at https://journals.biologists.com/dev/issue/152/20.

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
