## [Peer Review File · Development (Cambridge, England)]

***egal-1* and microtubules promote regeneration polarity in planarians**

Yochabed Miliard, Shannon Moreno, Lauren E. Cote and Peter W. Reddien

DOI: 10.1242/dev.204668

Editor: Kenneth Poss

Review timeline

Original submission: 21 January 2025

Editorial decision: 10 March 2025

First revision received: 30 July 2025

Accepted: 23 August 2025

Original submission

First decision letter

MS ID#: dev.204668

MS TITLE: *egal-1* and microtubules promote regeneration polarity in planarians

AUTHORS: Yochabed Miliard, Shannon Moreno, Lauren Cote and Peter W. Reddien

Dear Peter:

I apologize for the extreme delay in a decision -- we had been waiting for a 3rd review but decided to proceed. We have now received two referees' reports on the above manuscript, and have reached a decision. The referees' comments are appended below, or you can access them online: please go to:

As you will see, the referees express considerable interest in your work, but have some significant criticisms and recommend a substantial revision of your manuscript before we can consider publication. If you are able to revise the manuscript along the lines suggested, which may involve further experiments, I will be happy receive a revised version of the manuscript. Your revised paper will be re-reviewed by one or more of the original referees, and acceptance of your manuscript will depend on your addressing satisfactorily the reviewers' major concerns. Please also note that Development will normally permit only one round of major revision. If it would be helpful, you are welcome to contact us to discuss your revision in greater detail. Please send us a point-by-point response indicating your plans for addressing the referees' comments, and we will look over this and provide further guidance.

Please attend to all of the reviewers' comments and ensure that you clearly highlight all changes made in the revised manuscript. Please avoid using 'Tracked changes' in Word files as these are lost in PDF conversion. I should be grateful if you would also provide a point-by-point response detailing how you have dealt with the points raised by the reviewers in the 'Response to Reviewers' box. If you do not agree with any of their criticisms or suggestions please explain clearly why this is so.

Reviewer 1

Advance summary and potential significance to field

Milard et al. studied how *egal-1* and microtubules promote regeneration polarity in planarians. This work significantly advances ongoing efforts to identify the mechanism of blastema identity specification to match missing tissues. It represents a systematic exploration of wound-induced gene expression in longitudinal muscles. While not the pioneering study in this direction and lacking a strong phenotype, the research presents important findings for the field, as well as new antibody and additional resource datasets. My comments and suggestions aim to enhance the study's mechanistic insights and overall clarity. Additionally, the author should review the citations to ensure comprehensive inclusion of pertinent references. Critically, several items in the figures are absent from the text, which impedes the clarity and coherence of publication.

Comments for the author

1. In the abstract, the asymmetric expression of *notum* is not entirely unknown. It has been reported that *wnt1* and *wnt11-2* control the expression of *notum* in the posterior wound. Therefore, the statement in the abstract should be precise. Given that the final results did not report an outgrowth of anterior tissue at the posterior wound, it is important to test whether the combination RNAi of *wnt11-1* or *wnt11-2* RNAi together with *egal-1* would result in outgrowth.
2. Because the longitudinal muscle fibers are only ~200 μ m in length, comprehending the mechanism by which a dynein-binding protein and microtubule stability affect the asymmetric expression of *notum* poses a significant challenge. Could there be potential molecular mechanisms predictable within the model at cellular level across the body's length? For example, the diffusion hypothesis is comprehensible for proteins or secreted factors, but it is more difficult to imagine the similar behavior of mRNA within cells. Furthermore, would the suppression of dynein, dynactin, or kinesin result in a similar alteration of *notum* expression?
3. It must be an error to assert that $p_{adj} > 0.05$ in line 4 on page 6.
4. *Wnt1* expression is considered not dependent on longitudinal muscles. However, scRNA-seq data reveal high expression of *wnt-1* in these muscles. It is better to provide additional staining experiments to validate and compare it with previous findings.
5. I do not fully understand the statistical method described on page 7, "We analyzed the top 200 genes that were significantly wound-induced in the control dataset at 6 hpa Although not all of these differences in expression were statistically significant,.....". The authors analyzed the top 200 significantly wound-induced genes. However, they subsequently noted that not all of these expression differences reached statistical significance. Please clarify this apparent inconsistency.
6. The sequence of the Figure panels does not align with their order of referenced in the text, leading to confusion during reading. To enhance clarity, I recommend reorganizing the figures. Specifically, Figure 3A, B, and C, as well as Figure S9 should be reordered. Additionally, Figure S7 should be swapped with Figure S8. I also propose combining figures 3 and 4 into a single figure. When discussing the sequence similarity of *dd_6723* and *dd_7434*, it would be important to include their expression patterns to improve the clarity of the study on *dd_8302*.
7. It would be valuable to show whether the cluster of *notum*⁺ cells can be formed at the posterior pole at 3 dpa to clarify the expression of *notum* following the ectopic expression at the posterior wound.
8. It is unclear how *egal-1* and microtubule regulate the expression of *notum* in posterior facing longitudinal muscles. Further discussion on potential explanation within a single muscle fiber would be beneficial.
9. Is it possible that *egal-1* RNAi generates an injury stress, which could account for the increased levels of multiple gene expression level in Figure S9? FigS9B shows an obviously different gene expression between the colchicine treatment and controls, contrasting the interpretation in the text. Please verify this inconsistency and address it in the discussion.

10. On page 14, "Alternatively, in the absence of microtubules, some process could variably be disrupted such that in some animals a polarized response could still occur.indicating that animals ultimately correctly specified anterior-versus-posterior blastema fates despite defects in notum expression". I was unable to locate relevant results in the figures. Please clarify the data related to these statements.

11. Figure 6G and H and Figure 7 cannot be located within the text. These figures seem to be referenced in the Discussion section, yet they lack proper citation.

12. Several citations need to be included.

Nicolle A. Bonar, David Gittin, Christian P. Petersen. PMID: 35297964

Eric M. Hill, Christian P. Petersen. PMID: 29547123

Erik G. Schad, Christian P. Petersen. PMID: 31928872

Eric Hill and Christian P. Petersen. PMID: 27074666

Minor points:

1. The author list needs to label who is affiliated with the 5th address note.

In the Method section:

2. The concentration of dsRNA for feeding should be provided in detail.

3. What is the purpose of the mild shaking and light exposure on the third day after the last feeding?

In Figures and legends:

4. If I missed it, Figure 1D, Figure S1A, Figure S3A, Figure 2D, and Figure S5C are not referenced in the manuscript.

5. Fig3B and S8B lack scale bar length. Images at 18 hpa are without scale bar labels. The legends for Figure 3E and 3F need to be corrected.

6. There is an error in the labeling B and C in Figure 4, which does not correspond to the legend. Additionally, the images in Figure 4B lack scale bar labels.

7. Is Colchicine misspelled as Colchcine in Figure 6 B and 6C?

8. The format of supplementary figure legends requires improvement. For example, the one with Figure S7.

Reviewer 2

Advance summary and potential significance to field

In this study, Miliard et al. set out to identify genes and mechanisms controlling wound cell fate specification. The Reddien Lab has pioneered studies showing that wound-induced asymmetric notum expression in longitudinal muscle fibers oriented along the anterior-posterior axis is crucial for proper head-tail specification in planarian regeneration. A major question for regeneration fate decisions is how critical components are polarized along the muscle fibers. The authors used a scRNA-seq approach to recover differential gene expression in longitudinal muscle fibers along anterior and posterior wounds. The resulting data was compared to previous bulk RNA-seq results for myoD RNAi-treated planarian tail fragments. The identified genes were validated by expression analysis. One gene enriched in longitudinal muscles was dd_8302, which encodes an exonuclease domain-like protein with unclear homology, and they elected to name egalitarian-like 1 as it does share similarity with other egalitarian-like genes, especially in *C. elegans*. By expression analysis and generation of a polyclonal antibody to SMED-EGAL-1, the authors demonstrate EGAL-1 does localize to longitudinal fibers expressing notum. Disrupting egal-1 expression using RNAi does not cause gross regeneration phenotypes, but their analyses show that egal-1 RNAi disrupts asymmetric expression of notum. The transcript accumulates in posterior wounds. Finally, using pharmacological agents, the authors show that destabilizing microtubules also show accumulation of notum at posterior wounds. They conclude that egal-1 and microtubules are necessary for promoting regeneration polarity by regulating notum asymmetric expression in longitudinal fibers, but an exact mechanism is not well defined or explained. The study contains numerous elegant experiments that are well executed. Still, some of the results are preliminary or inconclusive, analyses lack sufficient explanation, and the connection between egal-1 and RNA transport or function is not established, diminishing the potential impact of the findings.

Comments for the author

Major Comments

The primary concern is that the lines of evidence between the role of *egal-1* and microtubule stability are not fully fleshed out or connected. The finding *egal-1* might be involved in RNA localization via active transport is not novel and is not clearly explained in the paper. How do the authors propose *egal-1* functions (i.e., what is the mechanism)? Figure 7 is vague and does not explicitly illustrate how the authors think the proteins act in the context of their results. Furthermore, as the authors are obviously aware of, other studies suggest that *notum* is expressed at all wounds, but it is well-established that *notum* expression is much more robust/persists at anterior wounds, and phenotypes have established that *notum* protein is inhibiting Wnt signaling at anterior wounds. In the context of *egal-1* RNAi, could the authors clarify how the loss of EGAL-1 impacts *notum* mRNA localization, or could it be stability or translation? Do they think EGAL-1 binds mRNAs, including *notum*, and transports them to anterior fiber ends? What is the mechanism?

- 1) P. 6, the methods do not explain how differential expression analysis was performed in Seurat and the settings. Is a fold change of >1 acceptable in this type of analysis?
- 2) The results and analysis support the selection of *egal-1*, but not all the images for the expression analysis in Figure 2C were sufficiently clear. It is difficult to glean the percent co-expression levels in the images of blastemas for *egal-1*, *dd_585*, and *dd_8461*, for example, and arrowheads indicating positive cells in the zoomed insets would be helpful. How were the cells counted (not discussed in methods), and what does the percentage represent?
- 3) Understanding the evolutionary relationship of *dd_8302* and the other planarian isoforms mentioned in the manuscript, which were named *egal-2* and *-3*, is critical for considering its potential role and conclusions. The analysis shown in Figure 3 could be more comprehensive. The authors generate a protein alignment with selected species. There is no phylogenetic analysis as suggested in the methods. Therefore, it is unclear whether *dd_8302* represents a bona fide *egalitarian-1* homolog and less so ortholog, considering it does not possess the Exo-C domain. Given that the domain is not only represented in the highly derived model ecdysozoans but is present in Cnidaria and Deuterostomia, it could be helpful to perform a proper phylogenetic analysis to assign homology to the planarian protein. Some programs employ algorithms for structural phylogenetic analysis that could be considered here.
- 4) The generation of the antibody is a significant accomplishment. It is challenging to generate antibodies for planarian proteins. Unfortunately, analysis of the utility or specificity was lacking. I agree the expression experiments and RNAi are convincing. But could the antibody cross-react with EGAL-2 and *-3*? What happens to the antibody staining if the paralogs are knocked down? There are also opportunities to examine where in muscle EGAL-1 is expressed using other markers like the 6G10 staining shown in Figure 6.
- 5) At least two instances in the paper suggest that additional studies will be necessary to understand the biochemical function of *egal-1*, or additional RNAi screens should provide insight into additional factors required for *egal-1* function. Having the polyclonal antibody on hand would be outstanding in exploring if the reagent could facilitate Mass Spec analysis or sequencing to identify bound targets. Could it work for those applications? Have the authors considered testing it?
- 6) Figure panels 6E-H are not mentioned in the narrative of the results. What is 6G10, and how was it used? If it marks muscle fibers, does it work in combination with anti-EGAL-1?
- 7) Figure 7 is beautiful, but the summary does not specifically summarize nor illustrate a mechanism by which the authors think *egal-1* functions in planarians.
- 8) The Discussion could discuss limitations on the analysis of *egal-1* function.

Minor comments

- 1) The title does not accurately reflect the study.
- 2) Whose current address is at Stanford?
- 3) As discussed throughout the comments, the methods lack details in some places (e.g., scRNA-seq differential expression analysis, cell counting, reagents, and procedures used in the study).

First revision

Author response to reviewers' comments

Reviewer 1: SUMMARY OF THE ADVANCE MADE IN THIS PAPER AND ITS POTENTIAL SIGNIFICANCE TO THE FIELD

Milard et al. studied how *egal-1* and microtubules promote regeneration polarity in planarians. This work significantly advances ongoing efforts to identify the mechanism of blastema identity specification to match missing tissues. It represents a systematic exploration of wound-induced gene expression in longitudinal muscles. While not the pioneering study in this direction and lacking a strong phenotype, the research presents important findings for the field, as well as new antibody and additional resource datasets. My comments and suggestions aim to enhance the study's mechanistic insights and overall clarity. Additionally, the author should review the citations to ensure comprehensive inclusion of pertinent references. Critically, several items in the figures are absent from the text, which impedes the clarity and coherence of publication.

Thank you for your efforts and comments.

SUGGESTIONS TO AUTHORS

1. In the abstract, the asymmetric expression of *notum* is not entirely unknown. It has been reported that *wnt1* and *wnt11-2* control the expression of *notum* in the posterior wound. Therefore, the statement in the abstract should be precise.

The abstract was edited to be more precise and to highlight what is unknown about the asymmetric activation of *notum* acting in polarized muscle fibers after injury occurs. We had to shrink the total word count in the abstract to fit journal requirements, restricting how much we could add to the abstract, but we go into detail on Wnts/Dsh/Activin in the intro and discussion. This study finds new processes involved in how properly oriented muscles can locally read out polarity in a way that is independent of muscle turnover, which is a novel contribution to the field of regeneration polarity. In the abstract, we now state: "... processes that occur within longitudinal muscle after injury ..." (the point being some emphasis on after wounding rather than the development of polarized fibers) and state these processes are "poorly understood" rather than unknown.

Given that the final results did not report an outgrowth of anterior tissue at the posterior wound, it is important to test whether the combination RNAi of *wnt11-1* or *wnt11-2* RNAi together with *egal-1* would result in outgrowth.

We tested whether combined RNAi of *wnt11-1*, *wnt11-2*, and *egal-1* would result in anterior tissue outgrowth at posterior-facing wounds and compared the results to either *egal-1* or *wnt11-1*, *wnt11-2* RNAi alone. We performed 18 days of RNAi feedings, then amputated the animals and waited for them to regenerate fully. Unfortunately, we did not observe any head outgrowth in the posterior of the animal, but did observe failure to regenerate posterior tissue as shown below. This phenotype is consistent with prior observations by Gitten et al. (2022). We used a shorter RNAi timeline and used a 1:1 dsRNA dilution in our experiment by design to seek a possible enhancement phenotype. Despite not observing enhancement of double-head formation, we naturally cannot rule out that such enhancement could be found with some combination of timing or dsRNA concentration, but it did not robustly emerge from the experiments we tried. Because *notum* expression at posterior-facing wounds occurs with higher frequency in other RNAi conditions than does posterior head formation (i.e., *activin-2* or *wnt11-2* RNAi), the balance of evidence suggests posterior-facing *notum* expression is not always sufficient for head formation. This points to the possibility of other inhibitory factors at posterior-facing wounds or other factors collaborating with *notum* at anterior-facing wounds to lead to the proper head-tail outcome, which we note in the text.

2. Because the longitudinal muscle fibers are only ~200 μ m in length, comprehending the mechanism by which a dynein-binding protein and microtubule stability affect the asymmetric expression of *notum* poses a significant challenge. Could there be potential molecular mechanisms predictable within the model at cellular level across the body's length? For example, the diffusion hypothesis is comprehensible for proteins or secreted factors, but it is more difficult to imagine the similar behavior of mRNA within cells.

We added the following text to the discussion to address how large-scale tissue polarity can in other contexts be connected to a local cellular polarity.

“The mechanism of polarity requires a local response in longitudinal fibers at any position across the anterior-posterior axis, and that the fibers are oriented in their response in the same direction. Some process likely sets this body-wide vector, possibly involving Wnt11 or other processes. Planar cell polarity is an example of a process that can generate polarity in cells with the same orientation across large length scales. Some conceptually similar process could in principle impact the coordinated polarity of longitudinal muscle fibers, however, planar cell polarity is not known to affect *notum* transcription but instead is implicated in cilia polarity in the planarian epidermis. How longitudinal muscle fibers each set their polarity to align to a body-wide vector is thus unknown, but we suggest that Egalitarian-1 and microtubules act within these polarized fibers to mediate a process occurring after injury that yields an asymmetric transcriptional response to injury“

Furthermore, would the suppression of dynein, dynactin, or kinesin result in a similar alteration of *notum* expression?

Based on our single cell sequencing data, there were 8 tubulin, 8 dynein, 12 kinesin, and 1 dynactin encoding genes with an expression level above 0.1 in longitudinal muscles. In response to this suggestion, we performed RNAi on all these components for 18 days, followed by amputation and assessment of *notum* expression. We did not find any of these RNAi conditions to result in ectopic *notum* expression at posterior-facing wounds at 18 hpa. There are several factors that could have precluded success with this approach, including the possibility of functional redundancy among these motor proteins or animal lethality associated with their depletion (which we observed for several genes) preventing animals surviving to the point where sufficient protein loss occurring to observe a defect in polarity. In the future, we can continue to explore the possibility of functional redundancy and continue to optimize variations on RNAi delivery approaches/schedules to minimize lethality or to optimize the timing of the assay with the timing of mortality.

3. It must be an error to assert that $p > 0.05$ in line 4 on page 6.

Thank you for noting this error, it has been adjusted accordingly.

4. Wnt1 expression is considered not dependent on longitudinal muscles. However, scRNA-seq data reveal high expression of *wnt-1* in these muscles. It is better to provide additional staining experiments to validate and compare it with previous findings.

We performed double FISH experiments using *wnt1* probes with longitudinal muscle markers

(*myoD* and *snail* probe pool) to assess the specificity of wound-induced *wnt1* expression to longitudinal muscle fibers (Supplementary fig 4). The wound area of three animals were counted and we found that 87.2% of *wnt1* expression co-localized with longitudinal muscle markers at 18 hpa. Counting was performed the same way as was performed with the other wound-induced genes (see Methods). This data is thus consistent with the scRNA-seq data. In prior work, *wnt1* expression was known to be highly specific to muscle with 96.6% co-expression with *collagen* at 16 hpa at anterior-facing wounds (Witchley et al, 2013). However, *myoD* RNAi conditions, which eliminate longitudinal muscle cells, did not robustly ablate *wnt1* expression at wounds at 6 hpa (Scimone et al. 2017). It is conceivable that there is a higher level of specificity to longitudinal muscle at 16hpa compared to 6 hpa. Regardless, we now note this higher than previously known/appreciated enrichment of *wnt1* expression in longitudinal muscle.

5. I do not fully understand the statistical method described on page 7, "We analyzed the top 200 genes that were significantly wound-induced in the control dataset at 6 hpa Although not all of these differences in expression were statistically significant,.....". The authors analyzed the top 200 significantly wound-induced genes. However, they subsequently noted that not all of these expression differences reached statistical significance. Please clarify this apparent inconsistency.

We revised the statistical method described on page 7 to make it clearer which two datasets are being compared.

6. The sequence of the Figure panels does not align with their order of referenced in the text, leading to confusion during reading. To enhance clarity, I recommend reorganizing the figures. Specifically, Figure 3A, B, and C, as well as Figure S9 should be reordered. Additionally, Figure S7 should be swapped with Figure S8. I also propose combining figures 3 and 4 into a single figure.

We revised the figure order so that the panels are now referenced in sequence with the text to improve clarity. Regarding the proposed combination of Figures 3 and 4, we considered this possibility carefully, and decided to keep them separate especially considering the new additions we made to Figure 4. In response to the second reviewer's comments, we added more analysis on the structural similarity of Egal-1 homologs in the supplement, and additional sub panels to Figure 4. Given these changes, we feel that presenting them separately best presents the data. Regarding Figure 4 we performed additional experiments with the anti-Egal-1 antibody. We showed colocalization of Egal-1 with the muscle fiber antibody 6G10 and we also showed association with microtubule fibers present in longitudinal muscle.

When discussing the sequence similarity of dd_6723 and dd_7434, it would be important to include their expression patterns to improve the clarity of the study on dd_8302.

We added FISH images and scRNA-Seq UMAP plots showing the expression patterns of *egal-2* (dd_6723) and *egal-3* (dd_7434) (Supplementary Figure 6). *egal-2* is expressed in the muscle whereas *egal-3* has sparse expression throughout. Neither gene appears to be wound-induced.

7. It would be valuable to show whether the cluster of *notum*⁺ cells can be formed at the posterior pole at 3 dpa to clarify the expression of *notum* following the ectopic expression at the posterior wound.

We added a figure to show that there was no formation of a *notum*⁺ pole at the posterior end of the tissue (Supplementary figure 12B, 13D) in either *egal-1* RNAi animals or colchicine-treated animals. We found that the anterior pole of colchicine-treated animals appeared more dispersed than the control and the animals did not regenerate (Supplementary figure 13D).

8. It is unclear how *egal-1* and microtubule regulate the expression of *notum* in posterior facing longitudinal muscles. Further discussion on potential explanation within a single muscle fiber would be beneficial.

We added text to the discussion to provide some potential explanation(s):

“At anterior-facing wounds, the anterior end of longitudinal fibers is removed, whereas the posterior end is removed at posterior-facing wounds. One possibility is that fibers are polarized such that an inhibitor of notum wound-induced expression is localized to the anterior side of the fiber, and either cannot be localized or is physically removed following injury (alternatively, a positive regulator of wound-induced notum could be localized to the posterior end of longitudinal fibers and be removed by injury at posterior-facing wounds). One scenario is that microtubules and Egalitarian-like are involved in the transport of some factor(s) (e.g., RNA or a protein) involved in this polarized process and that this factor is differentially affected by anterior- versus-posterior fiber end loss. RNAi of *egal-1* and microtubule inhibition leads to activation of notum at posterior-facing wounds, rather than loss of notum expression at anterior-facing wounds; this favors a model in which there exists an inhibitory process involving *Egal-1* and microtubules that is active in longitudinal fibers with a posterior injury (Fig. 7C). Molecular investigation of any possible polarized factors and their regulation by *Egalitarian* will be an important future direction to assess these possibilities.”

9. Is it possible that *egal-1* RNAi generates an injury stress, which could account for the increased levels of multiple gene expression level in Figure S9? FigS9B shows an obviously different gene expression between the colchicine treatment and controls, contrasting the interpretation in the text. Please verify this inconsistency and address it in the discussion.

We did not observe increased *notum*⁺ cells at anterior-facing wounds in *egal-1* RNAi animals compared to controls (Supplemental Figure 10B). This indicates that both anterior- and posterior- facing wounds did not equally elevate their expression levels and instead that the increase in *notum*⁺ cells at posterior-facing wounds is a specific defect. To further address this concern, we counted *wnt1*⁺ cells in *egal-1* RNAi conditions and did not observe elevated levels in *egal-1* RNAi conditions compared to controls (Supplemental Figure 11 B).

10. On page 14, "Alternatively, in the absence of microtubules, some process could variably be disrupted such that in some animals a polarized response could still occur.indicating that animals ultimately correctly specified anterior- versus-posterior blastema fates despite defects in notum expression". I was unable to locate relevant results in the figures. Please clarify the data related to these statements.

We added a FISH image of *notum*, *wntP-2* (posterior PCG), and *ndl-5* (anterior PCG) in colchicine-treated animals at 3 dpa that shows that despite disrupted *notum* expression, proper AP-fate specification still occurred (Supplementary Figure 13D).

11. Figure 6G and H and Figure 7 cannot be located within the text. These figures seem to be referenced in the Discussion section, yet they lack proper citation.

Thank you for noting this. We've added appropriate text references to every figure.

12. Several citations need to be included.

Nicolle A. Bonar, David Gittin, Christian P. Petersen. PMID: 35297964

Eric M. Hill, Christian P. Petersen. PMID: 29547123

Erik G. Schad, Christian P. Petersen. PMID: 31928872

Eric Hill and Christian P. Petersen. PMID: 27074666

These were added.

Minor points:

1. The author list needs to label who is affiliated with the 5th address note. **done**

In the Method section:

2. The concentration of dsRNA for feeding should be provided in detail. **done**

3. What is the purpose of the mild shaking and light exposure on the third day after the last feeding? **This has now been added to the methods.**

In Figures and legends:

4. If I missed it, Figure 1D, Figure S1A, Figure S3A, Figure 2D, and Figure S5C are not referenced in the manuscript.

All figures are now referenced in the manuscript

5. Fig3B and S8B lack scale bar length. Images at 18 hpa are without scale bar labels. The legends for Figure 3E and 3F need to be corrected. **done**
6. There is an error in the labeling B and C in Figure 4, which does not correspond to the legend. Additionally, the images in Figure 4B lack scale bar labels. **done**
7. Is Colchicine misspelled as Colchcine in Figure 6 B and 6C? **This has now been corrected, thankyou.**
8. The format of supplementary figure legends requires improvement. For example, the one with Figure S7. **done**

Reviewer 2: SUMMARY OF THE ADVANCE MADE IN THIS PAPER AND ITS POTENTIAL SIGNIFICANCE TO THE FIELD

In this study, Miliard et al. set out to identify genes and mechanisms controlling wound cell fate specification. The Reddien Lab has pioneered studies showing that wound-induced asymmetric notum expression in longitudinal muscle fibers oriented along the anterior-posterior axis is crucial for proper head-tail specification in planarian regeneration. A major question for regeneration fate decisions is how critical components are polarized along the muscle fibers. The authors used a scRNA-seq approach to recover differential gene expression in longitudinal muscle fibers along anterior and posterior wounds. The resulting data was compared to previous bulk RNA-seq results for myoD RNAi-treated planarian tail fragments. The identified genes were validated by expression analysis. One gene enriched in longitudinal muscles was dd_8302, which encodes an exonuclease domain-like protein with unclear homology, and they elected to name egalitarian-like 1 as it does share similarity with other egalitarian-like genes, especially in *C. elegans*. By expression analysis and generation of a polyclonal antibody to SMED-EGAL-1, the authors demonstrate EGAL-1 does localize to longitudinal fibers expressing notum. Disrupting *egal-1* expression using RNAi does not cause gross regeneration phenotypes, but their analyses show that *egal-1* RNAi disrupts asymmetric expression of notum. The transcript accumulates in posterior wounds. Finally, using pharmacological agents, the authors show that destabilizing microtubules also show accumulation of notum at posterior wounds. They conclude that *egal-1* and microtubules are necessary for promoting regeneration polarity by regulating notum asymmetric expression in longitudinal fibers, but an exact mechanism is not well defined or explained. The study contains numerous elegant experiments that are well executed. Still, some of the results are preliminary or inconclusive, analyses lack sufficient explanation, and the connection between *egal-1* and RNA transport or function is not established, diminishing the potential impact of the findings.

SUGGESTIONS TO AUTHORS

Major Comments

The primary concern is that the lines of evidence between the role of *egal-1* and microtubule stability are not fully fleshed out or connected. The finding *egal-1* might be involved in RNA localization via active transport is not novel and is not clearly explained in the paper. How do the authors propose *egal-1* functions (i.e., what is the mechanism)?

Figure 7 is vague and does not explicitly illustrate how the authors think the proteins act in the context of their results.

Furthermore, as the authors are obviously aware of, other studies suggest that notum is expressed at all wounds, but it is well-established that notum expression is much more robust/persists at anterior wounds, and phenotypes have established that notum protein is inhibiting Wnt signaling at anterior wounds. In the context of *egal-1* RNAi, could the authors clarify how the loss of EGAL-1 impacts notum mRNA localization, or could it be stability or translation? Do they think EGAL-1 binds mRNAs, including notum, and transports them to anterior fiber ends? What is the mechanism?

We added some text to the discussion to address possible mechanisms at play that could be investigated in the future. We hypothesize that *egal-1* and microtubules act upstream of *notum* transcription rather than regulating the post-transcriptional attributes of *notum* mRNA. One scenario is that Egal-1 binds and mediates the transport along microtubules of some mRNA for a *notum* regulator that is differentially affected by whether the injury removes the anterior or

posterior side of muscle fibers. We are hesitant to present too specific of a mechanism suggestion in the manuscript, such as in model figure format, because many possible mechanisms could be at play. However, we did add more to the text to describe possible mechanism(s):

“At anterior-facing wounds, the anterior end of longitudinal fibers is removed, whereas the posterior end is removed at posterior-facing wounds. One possibility is that fibers are polarized such that an inhibitor of notum wound-induced expression is localized to the anterior side of the fiber, and either cannot be localized or is physically removed following injury (alternatively, a positive regulator of wound-induced notum could be localized to the posterior end of longitudinal fibers and be removed by injury at posterior-facing wounds). One scenario is that microtubules and Egalitarian-like are involved in the transport of some factor(s) (e.g., RNA or a protein) involved in this polarized process and that this factor is differentially affected by anterior- versus-posterior fiber end loss. RNAi of *egal-1* and microtubule inhibition leads to activation of notum at posterior-facing wounds, rather than loss of notum expression at anterior-facing wounds; this favors a model in which there exists an inhibitory process involving *Egal-1* and microtubules that is active in longitudinal fibers with a posterior injury (Fig. 7C). Molecular investigation of any possible polarized factors and their regulation by *Egalitarian* will be an important future direction to assess these possibilities.”

1) P. 6, the methods do not explain how differential expression analysis was performed in Seurat and the settings. Is a fold change of >1 acceptable in this type of analysis?

We updated the Methods section to include additional details of the functions used for differential expression analysis (*FindMarkers* in Seurat for the cluster or sample origin), and for finding the average gene expression in a tissue cluster (*AverageExpression* function in Seurat). A log₂ fold change greater than 1 (>2 fold total difference) and a *padj* < 0.05 threshold was selected because most known muscle wound-induced genes fall within this range (Supplemental Data 1) and our aim was to capture a broad set of biologically relevant candidates that were then assessed further with FISH. We now note this reasoning in the text.

2) The results and analysis support the selection of *egal-1*, but not all the images for the expression analysis in Figure 2C were sufficiently clear. It is difficult to glean the percent co-expression levels in the images of blastemas for *egal-1*, *dd_585*, and *dd_8461*, for example, and arrowheads indicating positive cells in the zoomed insets would be helpful. How were the cells counted (not discussed in methods), and what does the percentage represent?

We added arrowheads in the zoomed insets of Figure 2C to highlight examples of double-positive cells for each gene. We also added the following to the methods to clarify how the cells were counted “To quantify double-positive cells for wound-induced gene expression in longitudinal muscle, cells positive for the candidate gene were first identified by viewing the corresponding fluorescence channel alone and determining that the fluorescence signal was around the cytoplasm of a single DAPI+ nucleus. The *myoD/snail* channel was then overlaid to assess co-localization.”

We also revised the Figure 2C legend for clarity. The reported percentage represents the number of cells co-expressing the wound-induced gene and the longitudinal muscle marker, divided by the total number of cells positive for the wound-induced gene.

3) Understanding the evolutionary relationship of *dd_8302* and the other planarian isoforms mentioned in the manuscript, which were named *egal-2* and *-3*, is critical for considering its potential role and conclusions. The analysis shown in Figure 3 could be more comprehensive. The authors generate a protein alignment with selected species. There is no phylogenetic analysis as suggested in the methods. Therefore, it is unclear whether *dd_8302* represents a bona fide *egalitarian-1* homolog and less so ortholog, considering it does not possess the Exo-C domain. Given that the domain is not only represented in the highly derived model ecdysozoans but is present in Cnidaria and Deuterostomia, it could be helpful to perform a proper phylogenetic analysis to assign homology to the planarian protein. Some programs employ algorithms for

structural phylogenetic analysis that could be considered here.

Because of variable lengths between folded domains observed in predicted structures of the N-terminal region among Egal-1 homologs across phyla and based on your suggestion, we decided to take a structure-based phylogenetic analysis approach to this topic. AlphaFold2 was used to predict structures of the top BLAST hits to Egal-1 in each representative species used. We then used Foldseek; using the *easy-search* function we generated pairwise structural similarity scores between candidate Egalitarian-family proteins (bit score value in Foldseek). These scores were normalized (1 - bits / maximum bits) and used to generate a dendrogram, providing a structure-based view of the evolutionary relationships of the similar proteins (Supplementary Figure 8B). Based on the dendrogram, *S. mediterranea* Egal-1 was more similar to Egal-like proteins in other Platyhelminthes (*S. polychroa* and *D. japonica*) than it was to Egal-2 and Egal-3, suggesting the divergence of these three paralogs occurred before the divergence of these other planarian species. These differences are also apparent by inspection in the structures shown in Supplementary Figure 8A. Well-characterized Egalitarian proteins in ecdysozoans (*Drosophila* and *C. elegans*) grouped closer to *S. mediterranea* Egal-1 than to the outgroup proteins used for this analysis that lack the conserved N-terminal domain but that have an ExoC exonuclease-like region (from cnidarians and deuterostomes). These analyses and new alphafold predictions further support the conclusion that Egalitarian-like proteins from protostomes form a distinct clade of related proteins that share a novel N-terminal domain and a C-terminal motif; there is variability in the nature of the central region in members of this family of proteins - there is an ExoC-like central domain in ecdysozoans and a unique fold in the central region of the spiralian Egalitarian-like proteins that we now uncover and show is similar across species.

4) The generation of the antibody is a significant accomplishment. It is challenging to generate antibodies for planarian proteins. Unfortunately, analysis of the utility or specificity was lacking. I agree the expression experiments and RNAi are convincing. But could the antibody cross-react with EGAL-2 and -3? What happens to the antibody staining if the paralogs are knocked down? There are also opportunities to examine where in muscle EGAL-1 is expressed using other markers like the 6G10 staining shown in Figure 6.

Thank you for the comments. We performed *egal-2* and *egal-3* RNAi and showed that the antibody is indeed specific to Egal-1. These results are now described in the Results section and included in new Supplementary Figure 9B. We also now show new staining that shows Egal-1 colocalizes with the 6G10 marker for muscle fibers; this data is added in Figure 4B. We also now show that there are foci of Egal-1 protein near nuclei of wounded longitudinal muscle and that these foci are lost with colchicine treatment. We also used the antibody together with a microtubule antibody to show that microtubule fibers are present within longitudinal fibers and associated with the Egal-1 protein and that Egal-1 protein projects along 6G10+fibers.

5) At least two instances in the paper suggest that additional studies will be necessary to understand the biochemical function of *egal-1*, or additional RNAi screens should provide insight into additional factors required for *egal-1* function. Having the polyclonal antibody on hand would be outstanding in exploring if the reagent could facilitate Mass Spec analysis or sequencing to identify bound targets. Could it work for those applications? Have the authors considered testing it?

We were also interested in identifying any potential binding partners to Egal-1 using the generated antibody and originally planned to do immunoprecipitation to identify Egal-1 binding partners. Unfortunately, we were unable to successfully identify the endogenous protein on a Western blot (see Western Blot image attached), despite forays into optimizing lysis and transfer conditions. Given this technical limitation that we could not follow the protein, we did not feel comfortable pursuing mass spectrometry-based identification of interactors at this time; it would take substantial time to optimize conditions and for follow-up on any possible hits. Additionally, we also attempted some RNA preparation from a pull-down of wounded lysates with the Egal-1N protein, but again were not confident we had worked out the technical conditions needed for actual pull-down with the *in vivo* protein. Whereas this does not exclude the possibility of future success with this approach, substantial optimization attempts would be needed, as well as follow-up endeavors on any possible hits.

6) Figure panels 6E-H are not mentioned in the narrative of the results. What is 6G10, and how was it used? If it marks muscle fibers, does it work in combination with anti-EGAL-1?

We updated the Results section to make sure all figure panels are referenced, we also added how we used the 6G10 antibody in the Methods section. The 6G10 antibody is a monoclonal antibody produced on formaldehyde-fixed cells from the planarian blastema and marks planarian muscle fibers in a manner similar to previous stainings of MHC-A, although the epitope is not mapped. This is now stated in the first mention of the antibody and in the methods. We added Figure 4B, where we used the 6G10 antibody in combination with the a-EGAL-1N antibody to show the localization of EGAL-1 in longitudinal muscle fibers.

7) Figure 7 is beautiful, but the summary does not specifically summarize nor illustrate a mechanism by which the authors think *egal-1* functions in planarians. We added some hypothesized mechanism discussion to the discussion section of the text and refer to this in the context of describing this figure and the paper findings. We decided to keep the figure itself restricted to conclusions that can be made at this point, and leave discussion to the text because anything we draw that goes further towards a very specific mechanism would just be one of multiple possibilities that cannot be distinguished between at present. However, in the text of the discussion we agree with your suggestion to add some of our thoughts, as this could help the reader visualize possibilities.

As noted above, we added the following text:

“At anterior-facing wounds, the anterior end of longitudinal fibers is removed, whereas the posterior end is removed at posterior-facing wounds. One possibility is that fibers are polarized such that an inhibitor of notum wound-induced expression is localized to the anterior side of the fiber, and either cannot be localized or is physically removed following injury (alternatively, a positive regulator of wound-induced notum could be localized to the posterior end of longitudinal fibers and be removed by injury at posterior-facing wounds). One scenario is that microtubules and Egalitarian-like are involved in the transport of some factor(s) (e.g., RNA or a protein) involved in this polarized process and that this factor is differentially affected by anterior- versus-posterior fiber end loss. RNAi of *egal-1* and microtubule inhibition leads to activation of notum at posterior-facing wounds, rather than loss of notum expression at anterior-facing wounds; this favors a model in which there exists an inhibitory process involving *Egal-1* and microtubules that is active in longitudinal fibers with a posterior injury (Fig. 7C). Molecular investigation of any possible polarized factors and their regulation by *Egalitarian* will be an important future direction to assess these possibilities.”

8) The Discussion could discuss limitations on the analysis of *egal-1* function. This is partially addressed in the above comment, where we hypothesize further in the discussion than we did in the original submission. We also note that important future paths include determining whether *Egal-1* binds RNA and what its binding partners and targets might be. One challenge we note for this is the difficulty of examining ends of single muscle fibers *in vivo*.

Minor comments

≈

1)The title does not accurately reflect the study.

We considered changing regeneration polarity to a statement about *notum* expression asymmetry but then felt that this would make it less accessible, and *notum* asymmetry is one of the main factors of regeneration polarity. We also considered adding information about the wound-induced genes but we felt that that would make the title too long. In the end we felt this title was succinct and hit the key points and would be readily interpretable to anyone reading the paper and we preferred it to alternatives that we considered.

2)Whose current address is at Stanford? Lauren Cote

3) As discussed throughout the comments, the methods lack details in some places (e.g., scRNA-seq differential expression analysis, cell counting, reagents, and procedures used in the study).

Thank you - we added more details to the text and methods.

Second decision letter

MS ID#: dev.204668R1

MS TITLE: egal-1 and microtubules promote regeneration polarity in planarians

AUTHORS: Yochabed Miliard, Shannon Moreno, Lauren Cote and Peter W. Reddien

Dear Peter,

I am happy to tell you that your manuscript has been accepted for publication in Development, pending our standard publication integrity checks.

Reviewer 1*Advance summary and potential significance to field*

The authors have addressed my comments. However, there are minor mistakes in the manuscript.

Comments for the author

1. "We assessed the expression of 139 genes from the scRNA-seq and myoD RNAi bulk RNA sequencing data in vivo with fluorescent in situ hybridization (FISH) at 18 hpa. 14 and 13 genes were shared between the and scRNA-seq datasets and the 6- and the 24-hour bulk RNA-seq, respectively."

How are the 139 genes in total counted from the prior paragraphs in the results?
"the" after "between" should be deleted.

2. The legend of Figure S2B are incomplete "Dot plot of expression of the top 30 genes enriched in the "wounded longitudinal muscle" cluster in different muscle types and"...

3. The legend of the supplementary figures should be carefully revised in the format and the content.

Reviewer 2*Advance summary and potential significance to field*

The authors' thoughtful responses to the critiques and careful revision, including the addition of new, beautiful data, have significantly improved mechanistic insights into the role of *egal-like-1* and microtubules in planarian regeneration polarity. I have a few minor comments and suggestions that I noted while reviewing the revision, listed below for the authors to consider.

Comments for the author

- 1) Page 6: The references for FTSTs should be added here, instead of citing Reddien (2022).
- 2) Page 12: I found the statement that "general responses to injury were not overtly impacted by *egal-1* RNAi (Fig. S11)" somewhat unclear. The authors could specify the time point, which I assume was 18 hpa. To this reviewer, it appears that all genes tested show a broader expression domain following *egal-1* RNAi, which is interesting and worth noting.
- 3) The authors have revised the Methods section in response to the review and added missing details. On page 23, the reference cited for 6G10 appears to be a review on the nervous system. The authors may wish to double-check that all references are correctly cited throughout the manuscript.